# GWAS identifies 14 loci for device-measured physical activity and sleep duration

Aiden Doherty [1,2,3,4], Karl Smith-Byrne[5], Teresa Ferreira[1,6], Michael V. Holmes[4,7,8], Chris Holmes[1,9], Sara L. Pulit[1,6,10,11] & Cecilia M. Lindgren[1,4,6,11]

Physical activity and sleep duration are established risk factors for many diseases, but their aetiology is poorly understood, partly due to relying on self-reported evidence. Here we report a genome-wide association study (GWAS) of device-measured physical activity and sleep duration in 91,105 UK Biobank participants, finding 14 significant loci (7 novel). These loci account for 0.06% of activity and 0.39% of sleep duration variation. Genome-wide estimates of ~15% phenotypic variation indicate high polygenicity. Heritability is higher in women than men for overall activity (23 vs. 20%, $p = 1.5 \times 10^{-4}$) and sedentary behaviours (18 vs. 15%, $p = 9.7 \times 10^{-4}$). Heritability partitioning, enrichment and pathway analyses indicate the central nervous system plays a role in activity behaviours. Two-sample Mendelian randomisation suggests that increased activity might causally lower diastolic blood pressure (beta mmHg/SD: −0.91, SE = 0.18, $p = 8.2 \times 10^{-7}$), and odds of hypertension (Odds ratio/SD: 0.84, SE = 0.03, $p = 4.9 \times 10^{-8}$). Our results advocate the value of physical activity for reducing blood pressure.

[1] Big Data Institute, Li Ka Shing Centre for Health Information and Discovery, University of Oxford, Oxford OX3 7LF, UK. [2] Nuffield Department of Population Health, BHF Centre of Research Excellence, University of Oxford, Oxford OX3 7LF, UK. [3] Institute of Biomedical Engineering, Department of Engineering Science, University of Oxford, Oxford OX3 7DQ, UK. [4] NIHR Oxford Biomedical Research Centre, Oxford University Hospitals NHS Foundation Trust, John Radcliffe Hospital, Oxford OX3 9DU, UK. [5] Cancer Epidemiology Unit, Nuffield Department of Population Health, University of Oxford, Oxford OX3 7LF, UK. [6] Wellcome Trust Centre for Human Genetics, University of Oxford, Oxford OX3 7BN, UK. [7] Clinical Trial Service Unit and Epidemiological Studies Unit, Nuffield Department of Population Health, University of Oxford, Oxford OX3 7LF, UK. [8] Medical Research Council Population Health Research Unit, Nuffield Department of Population Health, University of Oxford, Oxford OX3 7LF, UK. [9] Department of Statistics, University of Oxford, Oxford OX1 3LB, UK. [10] Department of Genetics, Center for Molecular Medicine, University Medical Center Utrecht, Utrecht 3584 CX, The Netherlands. [11] Program in Medical and Population Genetics, Broad Institute, Cambridge 02142 MA, USA. Correspondence and requests for materials should be addressed to A.D. (email: aiden.doherty@bdi.ox.ac.uk)

Physical inactivity is a global public health threat and is estimated to cost health-care systems ~$50 billion per year worldwide[1,2]. It is associated with a range of common diseases including multiple cardiometabolic outcomes such as obesity, type 2 diabetes, and cardiovascular diseases[3]. Alterations in sleep duration also associate with negative health outcomes, including cardiometabolic diseases[4] and psychiatric disorders[5]. In response, recent Canadian public health guidelines recommend how much physical activity and sleep duration youths should engage in for health benefit[6].

Twin and family studies have shown that self-reported daily physical activity is heritable, but with a large degree of heterogeneity in measurement methods and sample size ($h^2$ range: 0–78%)[7]. Recent GWAS studies have indicated a genetic contribution to physical activity, finding three variants[8] associated with standard UK Biobank activity metrics[9]. Recent GWAS have also reported common variants that are associated with sleep duration in UK Biobank[10–12]. However, GWAS of behavioural traits such as physical activity and sleep have mostly relied on self-reported data that are prone to measurement error and thus have limited statistical power. It is also possible that self-reported measures capture how well individuals perceive what they do, rather than what they actually do. Consequently, our understanding of the genes and biological pathways that underpin physical activity and sleep behaviours is still limited.

Here, we report a GWAS in 91,105 UK Biobank participants who were asked to wear activity trackers over a 7-day period. Our use of statistical machine learning, to derive objective measures of physical activity and sleep duration from raw device data, helps identify 14 genetic loci. The associated loci reveal pathways in the central nervous system that are associated with sleep duration and activity behaviours. We also show that physical activity might causally lower diastolic blood pressure and odds of hypertension.

## Results

**Statistical machine learning of behaviours from sensor data.**
We examined data from 101,307 UK Biobank participants who agreed to wear a wrist-worn accelerometer for 7 days[9] and additionally underwent genome-wide genotyping and imputation[13]. As part of the UK Biobank Accelerometer Working Group, we generated a continuous phenotype representing overall activity time[9] (Methods). Additionally, we applied a machine-learning model, using balanced random forests with Markov confusion matrices[14], to identify which one of four activity states {sleep, sedentary, walking, moderate intensity} an individual was in at any given time (Supplementary Data 1). We trained and evaluated the model in 153 free-living individuals (mean age = 42, female $n$ = 100) to distinguish between activity states, evaluated against reference wearable camera, time-use, and sleep diary information sources[14]. Our model achieved an overall classification score of kappa = 0.68 in correctly predicting what activity state an individual is in for any given 30-s time period (Supplementary Table 1). As detailed in a previous publication[14] this model was considered suitable for population inference in UK Biobank as the performance was not materially affected by age and sex characteristics (difference in kappa score: sex < 0.0001; age < 0.05). Therefore, our model was applied to over 100,000 UK Biobank participants (Fig. 1), where the machine-learned phenotypes demonstrated face validity[14] with clear and expected differences in sleep duration by self-reported chronotype status (Supplementary Fig. 1). These machine-learned phenotypes provided orthogonal information to the traditional overall activity time phenotype (Supplementary Fig. 2).

**Loci associated with physical activity and sleep duration.** After quality control, 91,105 participants of European descent and 9,926,106 single-nucleotide polymorphisms (SNPs) remained for subsequent genetic association analysis (Methods). To maximise statistical power, we performed our analyses using the BOLT-LMM software tool[15], which applies a linear mixed model to the data, allowing for inclusion of related individuals and those of varying genetic ancestry[16]. We additionally included assessment centre, genotyping array, age, age squared, and season of wear as covariates.

Our analysis identified 14 significant loci ($p < 5 \times 10^{-9}$) that were separated by at least 400 kilobases (kb) (Table 1, Fig. 1, and Supplementary Fig. 3). We empirically derived this threshold ($p < 5 \times 10^{-9}$) for genome-wide significance that considers multiple testing in densely imputed data[17]. Seven loci were novel and these include: one for overall activity (rs564819152 near *SKIDA1*, $p = 1.9 \times 10^{-9}$); two for sleep duration (rs2416963 near *MAPKAP1*, $p = 2.3 \times 10^{-10}$; and rs2006810 near *AUTS2*, $p = 3.9 \times 10^{-9}$); four for sedentary time (rs26579 near *MEF2C-AS2*, $p = 2.6 \times 10^{-9}$; rs25981 near *EFNA5*, $p = 3.0 \times 10^{-9}$; rs1858242 near *LOC105377146*, $p = 3.1 \times 10^{-9}$; and rs34858520 near *CALN1*,

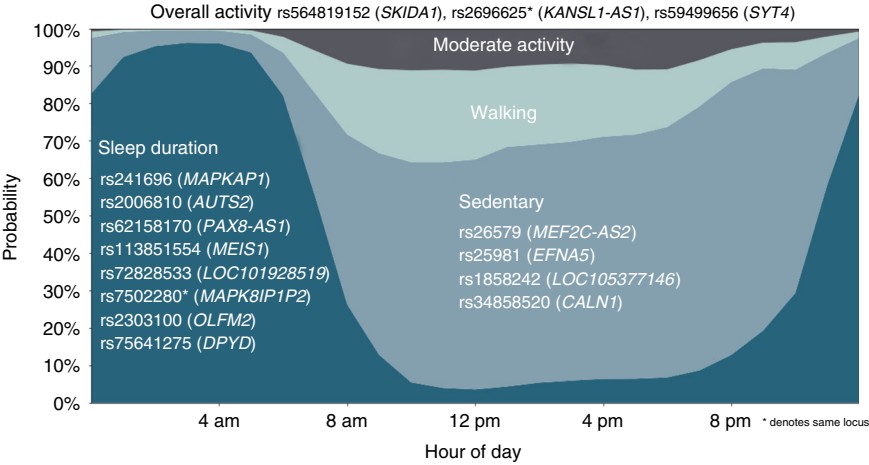

**Fig. 1** Genome-wide significant ($5 \times 10^{-9}$) loci associated with accelerometer-measured variation in sleep duration and physical activity behaviours in 91,105 UK Biobank participants

**Table 1 Genome-wide significant ($5 \times 10^{-9}$) loci associated with accelerometer-measured physical activity and sleep duration behaviours in 91,105 UK Biobank participants**

| Status | Trait | ID | Ch | SNP | Position | Nearest gene | Allele (effect/other) | EAF | Beta | SE | p |
|---|---|---|---|---|---|---|---|---|---|---|---|
| Novel | Overall activity | 1 | 10 | rs564819152 | 21,820,650 | SKIDA1 | A/G | 0.679 | 0.028 | 0.005 | 4.20E-09 |
| Novel | Sleep duration | 2 | 9 | rs2416963 | 128,241,414 | MAPKAP1 | C/T | 0.589 | 0.030 | 0.005 | 2.30E-10 |
| Novel | Sleep duration | 3 | 7 | rs2006810 | 69,902,152 | AUTS2 | T/C | 0.604 | -0.028 | 0.005 | 3.90E-09 |
| Novel | Sedentary | 4 | 5 | rs26579 | 87,985,295 | MEF2C-AS2 | G/C | 0.415 | 0.028 | 0.005 | 2.60E-09 |
| Novel | Sedentary | 5 | 5 | rs25981 | 106,822,908 | EFNA5 | G/C | 0.531 | 0.028 | 0.005 | 3.00E-09 |
| Novel | Sedentary | 6 | 3 | rs1858242 | 68,527,135 | LOC105377146 | A/G | 0.259 | 0.031 | 0.005 | 3.10E-09 |
| Novel | Sedentary | 7 | 7 | rs34858520 | 71,723,883 | CALN1 | A/G | 0.558 | 0.028 | 0.005 | 4.20E-09 |
| Known[8] | Overall activity | 8 | 17 | rs2696625 | 44,326,864 | KANSL1-AS1 | A/G | 0.77 | -0.037 | 0.005 | 3.20E-12 |
| Known[8] | Overall activity | 9 | 18 | rs59499656 | 40,768,309 | SYT4 | A/T | 0.655 | -0.028 | 0.005 | 1.90E-09 |
| Known Neale,[12,18,21,23,24] | Sleep duration | 10 | 2 | rs62158170 | 114,082,175 | PAX8-AS1 | A/G | 0.783 | -0.051 | 0.006 | 5.80E-20 |
| Known[18,23] | Sleep duration | 11 | 2 | rs113851554 | 66,750,564 | MEIS1 | G/T | 0.943 | 0.090 | 0.010 | 3.10E-18 |
| Known[23] | Sleep duration | 12 | 6 | rs72828533 | 19,065,680 | LOC101928519 | A/T | 0.818 | 0.043 | 0.006 | 2.70E-13 |
| Known[12,24] | Sleep duration | 8 | 17 | rs7502280 | 43,670,221 | MAPK8IP1P2 | T/G | 0.867 | 0.051 | 0.008 | 8.80E-11 |
| Known Neale,[21,24] | Sleep duration | 13 | 19 | rs2303100 | 9,968,434 | OLFM2 | C/T | 0.447 | -0.030 | 0.005 | 1.40E-10 |
| Known Neale,[23,24] | Sleep duration | 14 | 1 | rs75641275 | 98,327,133 | DPYD | A/C | 0.859 | 0.042 | 0.007 | 2.20E-10 |

Beta and SE are in standard deviation units
*EAF* effect allele frequency, *Ch* chromosome

$p = 4.2 \times 10^{-9}$); while no loci were identified for walking or moderate intensity activity (Table 1 and Supplementary Fig. 4). The remaining associations were known: two for overall and moderate intensity activity[8] (near *KANSL1-AS1* and *SYT4*) and six for sleep duration[12,18–22] (near *PAX8-AS1*, *MEIS1*, *LOC101928519*, *MAPK8IP1P2*, *OLFM2*, and *DPYD*). Setting genome-wide significance at the conventional threshold of $p < 5 \times 10^{-8}$ would have revealed 13 additional loci, 11 of which are novel (Supplementary Data 2).

We found concordance for loci previously reported using UK Biobank data for overall (near *PML*) and moderate intensity activity[8] (near *LINC02210-CRHR1, and SYT4*), sleep duration[12,19,23,24] (near *PUM1, CHCHD5, SAMMSON, BTBD9-AS1, LRP12, EBLN3P, BSX, KCNH5, PRKCB,* and *RPGRIP1L*), and sedentary time (near *LINC01930* and *LY6H*) (Supplementary Data 3). No other findings were confirmed from previous GWAS that relied on self-report measures of physical activity and sleep duration. To account for the likely non-linear U-shape relationship between sleep and disease outcomes[25], we ran additional GWAS on short and long sleep, defined as those in the bottom or top quintiles for sleep duration, respectively. We found three loci for short sleep (near *PAX8-AS1, MEIS1,* and *MAPK8IP1P2*) that were already identified for sleep duration. We found one locus for long sleep duration (near *PAX8-AS1*) that was also already identified for sleep duration. This indicates that our sleep duration results are not driven by individuals who are very short or long sleepers.

Genomic control lambda across the GWAS ranged from 1.1 to 1.20, indicating modest deviation in the test statistics compared with the expectation. Estimation of the LD Score intercepts, using LD score regression[26], revealed intercepts near 1.00 (range: 0.98–1.02), indicating that the genomic inflation was due to a polygenic architecture rather than inflation due to uncorrected population structure. We performed approximate conditional analysis in each locus reaching genome-wide significance using Genome-wide Complex Trait Analysis[27] (GCTA) but did not identify secondary signal in any locus. Gene-based analysis using FUMA[28], with input SNPs mapped by position to 18,232 protein-coding genes, identified an additional locus for walking (near *HP1BP3*) and moderate activity (near *KCNA6*) (Supplementary Data 4).

**Genetic architecture of physical activity and sleep duration.** Using heritability estimates from BOLT-LMM[15] (Methods), we found all traits to have modest heritability, ranging from 10% for moderate intensity activity to 21% for overall activity (Supplementary Fig. 3). Genome-wide significant loci account for 0.06% (overall activity) to 0.33% (sleep duration) of phenotypic variance across the traits (Supplementary Fig. 3). Additional analyses testing for polygenic trait architecture indicate that SNPs with *p*-values above genome-wide significance add significantly to the phenotypic variance explained. For example, explained phenotypic variance ranged from 12.9% for sedentary behaviour to 18.1% for sleep duration when we selected independent SNPs ($r^2 < 0.1$) with $p < 5 \times 10^{-3}$ and distance > 250 kb from index SNPs.

We applied partitioned heritability[29] analysis for tissue and functional categories using LD score regression[26]. We identified significant tissue enrichments ($p < 1 \times 10^{-3}$, accounting for ten tissue types and five traits) in the central nervous system for all traits apart from moderate activity. We also identified significant enrichments for adrenal/pancreatic and skeletal muscle tissues for overall activity and sleep duration traits (Supplementary Fig. 5). Regions of the genome annotated as conserved across mammals were enriched ($p < 4.7 \times 10^{-4}$, accounting for 21 functional categories and five traits) for overall activity, sleep duration, and sedentary behaviour traits. These findings support a role for physical activity and sleep behaviours throughout mammalian evolution.

For all sleep duration and activity phenotypes, we used FINEMAP[30] to identify credible sets of causal SNPs (meeting a $\log_{10}$ Bayes Factor > 2) in a 1 Mb window around each index SNP, on the assumption that there is a maximum of five causal variants per locus. Three loci contained plausible causal variants in regions spanning < 15 kb (Supplementary Table 2). One locus contained a single plausible causal variant (*rs113851554*) near *MEIS1* in chromosome 2 for sleep duration.

**Sexual dimorphism in physical activity and sleep duration.** Given that known sex differences exist for physical activity levels[9] and muscle and fat mass distribution[31], we investigated genetic sexual dimorphisms in our activity traits. Genetic correlations[32]

of trait architectures as measured using LD score regression[26] showed strong correlation between men and women; correlations ranged from 0.92 for overall activity to 0.97 for sleep duration (Supplementary Fig. 6). In addition, there was no evidence of sexual dimorphism using strict criteria for heterogeneity of effects between men and women ($p < 5 \times 10^{-9}$).

Heritability in women was higher than that in men for overall activity (23% vs. 20%, $p = 1.5 \times 10^{-4}$) and sedentary behaviours (18% vs. 15%, $p = 9.7 \times 10^{-4}$) (Supplementary Fig. 6). We found no evidence for sexually dimorphic heritability in sleep duration ($p = 0.19$), walking ($p = 0.54$), and moderate intensity activities ($p = 0.69$). We found some evidence of sexually dimorphic variance explained in 4 of the 5 traits, likely due to the greater sample size of female participants (Supplementary Fig. 6).

**Association of physical activity and sleep with other traits.** While physical activity and sleep duration are established risk factors for multiple diseases from observational epidemiology, the extent to which their underlying genetic architectures are shared with disease phenotypes is unknown[7]. We conducted various analyses that included: (1) genome-wide genetic correlations between our traits and those from publicly available GWAS data using LD score regression[26] on the LD-Hub web resource[33]; (2) phenome-wide association (pheWAS) analysis for index SNPs using the Oxford Brain Imaging Genetics Server[34] (big.stats.ox.ac.uk); and (3) a lookup of the NHGRI GWAS catalogue[35] for related SNPs associated with other diseases and traits (within 400 kb and $r^2 > 0.2$). Activity and sleep traits showed genetic associations with many other independent traits in LD score regression ($n = 155$), pheWAS ($n = 71$), and GWAS catalogue lookup ($n = 11$) analyses (Supplementary Data 5–7). Associations with anthropometric and cognitive health traits were particularly common (Supplementary Note 1). In general, increases in overall activity and walking phenotypes were genetically correlated with improved health status, while increases in sleep duration were negatively correlated with fluid intelligence scores and health status. Increases in sedentary time were genetically correlated with increased fluid intelligence scores, but decreased health status (Supplementary Note 1 and Supplementary Data 5).

To investigate whether activity and sleep might contribute causally to disease outcomes, we performed Mendelian randomisation (MR) analysis in 278,374 UK Biobank participants who were not included in our discovery analysis, and other publicly available GWAS summary data from the MR-Base web platform[36,37] (Methods). Rather than selecting all significant traits from the previous genetic correlation analyses, we decided to only focus on major diseases and risk factors. These included cardiovascular disease (CHD, stroke, heart failure), type 2 diabetes, Alzheimer's, cancer, blood pressure, and anthropometric traits (BMI and body fat %). For the MR analysis of each trait, we selected instrument variables at loci with $p < 5 \times 10^{-6}$, as they explain more phenotypic variance than traditional genome-wide loci ($p < 5 \times 10^{-8}$) with negligible effects on horizontal pleiotropy (Supplementary Fig. 7). We then followed a number of steps to denote potential causality with disease outcomes (Methods, Supplementary Note 2). These included: maximum likelihood estimates[38], leave-one-out analyses, robustness against classic confounders and pleiotropy via median and mode estimators, MR-Egger, and bi-directional tests[39]. We found consistent evidence of inverse relationships for overall activity with body mass index (beta SD of BMI per SD higher overall activity: −0.14, SE = 0.015, $p = 8.7 \times 10^{-20}$), diastolic blood pressure (beta mm Hg per SD: −0.91, SE = 0.18, $p = 8.2 \times 10^{-7}$), and hypertension (Odds ratio per SD: 0.84, SE = 0.03, $p = 4.9 \times 10^{-8}$). (Supplementary Data 8–10 and Supplementary Fig. 8). None of these

results appear affected by horizontal pleiotropy or a single variant or association with classic confounders (Supplementary Data 11). However, there was evidence of a bi-directional relationship between adiposity and overall activity (Supplementary Data 12), which is considered plausible[40,41] despite some evidence of horizontal pleiotropy.

**Potential functional and biological mechanisms.** We anticipated the use of objective measures would not only identify activity and sleep associated variants, but also provide insight on potential functional and biological mechanisms for these traits. First, we used DEPICT[42] at suggestive loci with $p < 1 \times 10^{-5}$ to identify gene sets enriched for accelerometer-defined phenotype associations; and tissues and cell types in which genes from associated loci are highly expressed. This analysis did not yield any significant results (FDR < 0.05). Secondly, we used the FUMA[28] web platform to identify tissues and gene pathways enriched for genetic signals (Methods). Here, we found brain tissues to be enriched ($p < 1.8 \times 10^{-4}$), in particular at the cerebellum, frontal cortex, brain cortex, and anterior cingulate cortex for at least two accelerometer traits (Supplementary Fig. 9). We also identified pathways enriched in neurological disease, brain structure, and cognitive function, among other traits (Supplementary Fig. 10).

**Discussion**

Our analysis of 91,105 individuals identified 14 loci significantly associated with objectively measured physical activity and sleep traits. While these loci alone account for less than 0.39% of phenotypic variation, our analyses suggest that as much as 18% of physical activity and sleep duration variation might be accounted for by common (minor allele frequency > 1%) genetic variation. We found heritability was higher in women than in men for overall activity and sedentary behaviour, but no differences were found for sleep, walking, and moderate intensity activity time. As anticipated, physical activity and sleep duration have polygenic architectures and share biology with multiple traits including intelligence, education, obesity, and cardiometabolic diseases.

Our results advocate the value of physical activity for the reduction of blood pressure. Until now, it has been difficult to discern whether associations between activity and disease are truly causal or biased due to reverse causality, confounding, and self-report measurement error associated with traditional observational studies[43]. Ascertainment bias is also an issue in randomised controlled trials of physical activity, due to difficulties in blinding participants to behavioural interventions[44]. Debate on whether obesity is a determinant of physical activity, or vice-versa, has previously lacked information due to a paucity of instrument variables for physical activity[7,40]. Our MR analyses indicate that activity and adiposity share a bi-directional relationship. In summary, our MR analyses suggest that higher levels of physical activity might causally lower diastolic blood pressure, and odds of hypertension.

We observed an important role for the central nervous system with respect to physical activity and sleep, consistent with other studies that have used UK Biobank data[8]. In addition to these findings from heritability partitioning, enrichment and pathway analyses; our gene-set and pheWAS analyses showed genetic overlap with neurodegenerative diseases, mental health wellbeing, and brain structure. Intriguingly, a recent GWAS of BMI highlighted a key role for neurological pathways in overall fat distribution[45], while a GWAS of fat distribution as measured by waist-to-hip ratio highlight adipose pathways[46]. It is, therefore, possible that our findings might be driven through obesity, as we observed strong genetic correlations between activity and obesity

traits, and the strength of association for some activity loci was attenuated when including BMI as an additional covariate. However, we have likely only begun to understand the complex interplay between activity, cardiometabolic phenotypes, and neurologic health. Animal models might be informative to answer such questions, as our heritability partitioning analysis supports a role for physical activity and sleep behaviours in regions conserved in mammals.

Our work demonstrates the utility of device measures to identify genetic associations with activity phenotypes. For example, we found that previous self-reported literature has significantly underestimated heritability for physical activity[8] (~ 5% vs. ~ 20% for accelerometer-measured traits). The larger-sample sizes that are currently available for self-reported phenotypes results in more identified loci[8], however, heritability and potential explained variance appears higher for the objective traits. Recall, comprehension, and social-desirability bias are prevalent in the measurement of behavioural traits such as physical activity, where phenotypic agreement is generally poor between subjective and objective measures (typically $r < 0.2$)[47]. It is likely that combination of self-report and objective measures will be needed to further our understanding of the genes and biological pathways that underpin physical activity.

This study has many strengths including: objectively measured behavioural phenotypes validated in free-living environments, use of linear mixed models, and large sample sizes all key to maximise power in genetic discovery analyses; and Mendelian Randomisation analyses, for delineating causal relationships. Given the cohort age range (45–80 years) and inclusion of European ancestry individuals only, we cannot necessarily assume our results generalise to other age groups or ancestral populations. Therefore, replication of our findings in non-European populations with both genetic and accelerometer-measures of sleep duration and physical activity will be important in the future. The high polygenicity of these traits makes it difficult to precisely estimate replication sample sizes. Assuming the replication $p$-value is set at $0.05/15$ loci $= 3.3 \times 10^{-3}$, and variance explained by a SNP ranges between $1.5–5.0 \times 10^{-4}$, we estimate between ~ 36 and ~ 119 k participants are necessary to provide 90% power to replicate the loci listed in Table 1. Replication in other cohorts would also help refine estimates of phenotypic variance explained by genome-wide significant loci, which are currently calculated in the discovery dataset reported in this paper. To account for the likely U-shape relationship between sleep and disease outcomes[25], rather than a linear relationship as assumed in current models, future work will be also required for MR sleep analysis. To develop machine-learned models of sleep duration in free-living environments, we did not use traditional cumbersome polysomnography methods, and instead relied on self-reported sleep diaries which can be prone to measurement error. While the participants in our machine learning free-living training dataset are not a random subsample of the UK Biobank study, the size of our training dataset has helped provide a diverse set of representative behaviours, rather than individuals, which is important for model development[14].

In summary, our GWAS of device-measured physical activity and sleep duration has identified 14 associated genetic loci. Our analysis shows shared genetic pathways with multiple traits including intelligence-related phenotypes and cardiometabolic disease. In support of national and international clinical guidance, this study provides strong evidence for physical activity in the prevention of blood pressure.

## Methods

**Participants**. Study participants were from the UK Biobank[13] where a subset of 103,712 participants agreed to wear a wrist-worn accelerometer for a 7-day-period

between 2013 and 2015[9]. Participants who were slightly more likely to consent included: women, those aged 55–74 years, those with higher socio-economic status, better physical health status, and less time since the baseline assessment (all odds ratios < 1.1). There were no significant differences in consent by self-reported physical activity status. This study (UK Biobank project #9126) was covered by the general ethical approval for UK Biobank studies from the NHS National Research Ethics Service on 17th June 2011 (Ref 11/NW/0382). For the development and free-living evaluation of accelerometer machine-learning methods, we used a validation set of 153 participants recruited to the CAPTURE-24 and ENERGY-24 studies where adults aged 18–91 were recruited from the Oxford region in 2014–2015[14,48]. Participants were asked to wear a wrist-worn accelerometer (same as in UK Biobank) for a 24-h period and then given a £20 voucher for taking part in this study that received ethical approval from University of Oxford (Inter-Divisional Research Ethics Committee (IDREC) reference number: SSD/CUR-EC1A/13–262).

**Device**. Participants wore an Axivity AX3 wrist-worn triaxial accelerometer on their dominant hand at all times over a 7-day-period. It was set to capture triaxial acceleration data at 100 Hz with a dynamic range of + −8g. This device has demonstrated equivalent output[49] to the GENEActiv accelerometer, which has been validated using both standard laboratory and free-living energy expenditure assessment methods[50,51]. For data pre-processing we followed procedures that we developed as part of the UK Biobank accelerometer data processing expert group[9], that included device calibration, resampling to 100 Hz, and removal of noise and gravity.

**Development and validation of machine-learned phenotypes**. To create a reference set of labels for sleep, sitting/standing, and walking time, participants in the validation study also wore cameras in natural free-living, rather than constrained laboratory, environments. Wearable cameras automatically take photographs every ~ 20 s, have up to 16 h battery life and storage capacity for over 1 week's worth of images. When worn, the camera is reasonably close to the wearer's eye line and has a wide-angle lens to capture everything within the wearer's view. Each image is time-stamped so duration of sedentary behaviour, waking, and a range of other physical activity behaviours can be captured[52]. To extract sleep information, participants were asked to complete a simple sleep diary, as used in the Whitehall study, which consisted of two questions[53]: '*what time did you first fall asleep last night?*' and '*what time did you wake up today (eyes open, ready to get up)?*'. Participants were also asked to complete a Harmonised European Time Use Survey diary[54], and sleep information from here was extracted in cases where data was missing from the simple sleep diary. An overview of the distribution of activity behaviours in this training dataset is provided in Supplementary Fig. 11.

For every non-overlapping 30-s time window, we then extracted a 126-dimensional feature vector representing a range of time and frequency domain features[14]. For activity classification we used random forests[55] which offer a powerful nonparametric discriminative method for classification that offers state-of-the-art performance[56]. Random forests are able to classify data points, but do not have an understanding of our data as having come from a time series. Therefore we then used a hidden Markov model[57] (HMM) to smooth our predictions. This smoothing corrects erroneous predictions from the random forest, such as where the error is a blip of one activity surrounded by another and the transitions between those two classes of activity are rare. This allowed us to train a model using all free-living ground truth data, achieving 79% accuracy (kappa = 0.68) across classes of interest[14] (Supplementary Table 1).

**Physical activity and sleep duration in UK Biobank**. To predict sleep, sedentary, walking, and moderate intensity activity time, we applied our machine-learning method to predict behaviour for each 30-s epoch in 103,712 UK Biobank participants' accelerometer data. A sedentary behaviour is one which has a MET (Metabolic Equivalent of Task) energy expenditure score of ≤ 1.5 and occurs in a sitting, lying, or reclining posture[58]. As an exception to this rule we also categorised the following annotations as sedentary behaviour: driving (code 16010) where assigned MET score is 2.5, and some instances of non-desk work (code 21010 rather than 5080). The complete list of image derived annotations and the phenotype class mappings are available in Supplementary Data 1. For any given time window (e.g., 1 h, 1 day, etc.) the probability of a participant engaging in a specific behaviour type was expressed as the number-of-epoch-predictions-for-class divided by the number-of-epochs. For overall activity levels, we selected average vector magnitude for each 30-s epoch, which is the recommended variable for activity analysis[9]. This variable has been shown to account for 44–47% explained variance vs. combined sensing heart-rate + trunk-acceleration (a proxy for free-living physical activity energy expenditure) in 1695 UK adults[51].

Device non-wear-time was automatically identified as consecutive stationary episodes lasting for at least 60 min[9]. These non-wear segments of data were imputed with the average of similar time-of-day data points, for each behaviour prediction, from different days of the measurement. We excluded participants whose data could not be calibrated ($n = 0$), had too many clipped values ($n = 3$), had unrealistically high values (average vector magnitude > 100 mg) ($n = 32$), or

who had poor wear-time ($n = 6860$). We defined minimum wear-time criteria as having at least 3 days (72 h) of data and also data in each 1-h period of the 24-h cycle[9].

**Genotyping and quality control**. The UK Biobank provides ~ 92 million variants, including imputation based on UK10K haplotype, 1000 Genomes Phase 3, and Haplotype Reference Consortium (HRC) reference panels[59]. We excluded SNPs with MAF < 0.1% (~ 83 M) and imputation $R^2 < 0.3$ (~9 k). We removed participants who self-reported as being non-white ($n = 3192$), had abnormal genetic versus self-reported sex mismatches ($n = 36$) or sex chromosome aneuploidy ($n = 71$). This left 91,105 participants of European descent (39,968 men; 51,137 women), and 9,926,106 SNPs, for subsequent genetic association analysis.

**Identifying loci associated with physical activity and sleep**. To improve our power to detect associations, we used BOLT-LMM[15] to perform linear mixed models analysis. As this adjusts for population structure and relatedness, we could include many additional individuals ($n = 13,320$) than if following the common practice of analysing a reduced set of unrelated white British individuals using linear regression[60]. We included assessment centre, genotyping array, age, age squared, and season as covariates. Sensitivity analysis using LD score regression[26] on the overall activity trait showed we did not need to include principal components of ancestry as covariates in our linear mixed models (lambda = 1.20 vs. 1.20, LD intercept = 1.0085 vs. 1.0092, genome-wide genetic correlation ($R_g$) = 0.99). We decided against including too many covariates to avoid the introduction of any possible collider bias that could have an effect on downstream analyses. In addition, we found little difference with the inclusion of: (1) sex as an additional covariate ($R_G = 0.99$); (2) BMI as an additional covariate ($R_G = 0.93$); and (3) both sex and BMI as additional covariates ($R_G = 0.93$). Supplementary Data 2 reports a sensitivity analysis for each locus with sex and BMI included as additional covariates, where all attenuations are driven by BMI alone. These results should be cautiously interpreted due to the potential for collider bias when adjusting for BMI.

We used PLINK[61] (version 1.9) to exclude candidate variants ($p < 1 \times 10^{-5}$) that deviated from Hardy-Weinberg equilibrium ($p < 1 \times 10^{-7}$). To identify genome-wide-significant (GWS) loci, we defined a distance criterion of + −400 kb surrounding each GWS peak ($p < 5 \times 10^{-9}$). We empirically derived this threshold for genome-wide significance that considers multiple testing and densely imputed data[17]. For reporting, we also identified loci using a traditional GWS value of $p < 5 \times 10^{-8}$.

To identify novel loci, we considered index SNPs falling outside 400 kb of a SNP previously associated with one of the following: self-report sleep duration or physical activity from the NHGRI-EBI GWAS catalogue[35], CHARGE[21] and CARe[62] consortia; an analysis of our accelerometer-measured overall and moderate intensity activity variables[9] with loci reported in 91,000 UK Biobank participants[8]; accelerometer-measured sleep duration genes tested for replication in recent bioRxiv papers that used ~ 85,000 UK Biobank participants[22,23]; self-reported overall activity loci reported in bioRxiv papers using ~ 377,000 and ~ 120,000 UK Biobank participants, respectively[8,63]; self-reported sleep duration GWAS results in recent bioRxiv papers that used up to ~ 446,000 UK Biobank participants[12,24]; and sleep duration (field #1160), television watching time as a proxy for sitting (field #1070), and walking (field #874) GWAS results in ~ 330,000 UK Biobank participants published by the Neale lab at the Broad Institute.

To identify additional signals in regions of association, we performed approximate joint and conditional SNP association analysis in each locus using the Genome-wide Complex Trait Analysis[27] (GCTA) tool. Any lead SNPs identified in a known high-LD area[64] between 43.5 and 45.5 mb at chromosome 17 were treated as a single large locus in GCTA analysis.

Gene-based analysis were performed with MAGMA[65] v1.6, as implemented on the FUMA[28] web platform. Input SNPs were mapped by position to genes obtained from 18,232 protein-coding genes obtained via Ensembl[66] build 85. Bonferroni-correction was used to define genome-wide significance. To identify novel loci, we looked for genes more than 400 kb away from loci identified in the above SNP association analysis.

**Genetic architecture of physical activity and sleep duration**. To estimate heritability of each trait, we used BOLT-LMM[15] for computational efficiency, after sensitivity analysis showed negligible differences in estimates between BOLT-LMM and restricted maximum likelihood analysis[67] implemented using the BOLT-REML software tool. We applied partitioned heritability[29] analysis across the phenotypes by tissue and functional category using LD score regression[26] with the LDSC tool. Significant enrichments for individual traits were identified using thresholds of $p < 1 \times 10^{-3}$ for tissues (i.e., $p < 0.05/10$ cell types/5 traits) and $p < 3.57 \times 10^{-4}$ for functional categories (i.e., $p < 0.05/28$ categories/5 traits). We also investigated enrichments in the median value across all accelerometer-measured traits.

To calculate the explained phenotypic variance for each trait, we used the PRSice[68] tool to generate polygenic risk scores for the lead SNP in GWS loci and also for all SNPs with $p < 5 \times 10^{-3}$, distance > 250 kb from index SNPs, and $r^2 < 0.1$. The same participant inclusion criteria were used as for association analysis.

To identify plausible causal SNPs associated with sleep and activity phenotypes, we used the FINEMAP[30] software. Configurations of plausible causal SNPs from a

1 Mb window around each genome-wide significant locus were calculated on the assumption that there were a maximum of five causal variants per locus. Across all loci, we defined plausible causal SNPs as those meeting a $\log_{10}$ Bayes Factor > 2.

**Sexual dimorphism in physical activity and sleep duration**. To investigate potential sources of sex heterogeneity, we ran the aforementioned genome-wide association analyses for men and women independently. Genetic correlations[32] of trait architectures between men and women were measured using LD score regression[26] for each phenotype. We also tested for heterogeneity of effect estimates[31] between men and women for all SNPs using the EasyStrata[69] tool, with $p < 5 \times 10^{-9}$ to assess significance. Heritability differences between men and women for each trait were assessed by extracting a two-tailed $p$-value from the $z$-score in equation (1), where var indicates variance.

$$z = \frac{h^2_{females} - h^2_{males}}{\sqrt{\text{var } h^2_{females} - \text{var } h^2_{males}}} \tag{1}$$

To estimate genetic contributions to phenotypic variance for men and women separately, we selected index SNPs from loci with a less stringent $p < 5 \times 10^{-7}$, due to smaller sample sizes within strata.

**Association of physical activity and sleep with other traits**. To estimate genetic correlations between our accelerometer-defined phenotypes and other complex traits and diseases, we used LD score regression[26] implemented in the LD-Hub web resource[33]. To assess significance, we corrected for 4160 tests (5 traits x 832 phenotypes available on LD-Hub) with $p < 1.2 \times 10^{-5}$.

To examine whether the genome-wide significant SNPs identified in our analysis affected other traits, we used the Oxford Brain Imaging Genetics Server (big.stats.ox. ac.uk[34]) to perform a phenome-wide association study (PheWAS) on almost 4000 traits in UK Biobank participants. This included GWAS results of ~ 2000 phenotypes in ~ 330,000 UK Biobank participants published by the Neale lab at the Broad Institute, and the remaining traits were brain imaging-derived phenotypes measured on a subset of ~ 10,000 UK Biobank participants[34]. To assess significance, we corrected for 40,000 tests (10 loci x 4000 traits) with $p < 1.25 \times 10^{-6}$. Additionally, we extracted previously reported GWAS associations within 400 kb and $r^2 > 0.2$ of accelerometer index SNPs from the NHGRI GWAS catalogue[35].

To investigate whether activity and sleep might causally contribute to disease outcomes, we performed Mendelian Randomisation (MR) analysis. Rather than selecting all significant traits identified in other correlation analysis, we decided to concentrate on major diseases and peripheral risk factors. These included vascular disease (CHD, stroke, heart failure), diabetes, Alzheimer's, major cancer subtypes, blood pressure, and anthropometric traits (BMI and body fat %). Disease phenotypes were prepared following similar procedures as used for UK Biobank variables in the LD-Hub web resource[33]. We defined hypertensive cases as individuals with systolic blood pressure of > 140 mm Hg, or a diastolic blood pressure of > 90 mm Hg, or the report of blood pressure medication usage. For the analysis of systolic and diastolic blood pressure, we corrected blood pressure measures in people on antihypertensive drugs by adding 15 mm Hg to systolic and 10 mm Hg to diastolic blood pressure, in keeping with the approach taken by genome-wide association studies[70]. Similar to the LD-Hub web resource[33], we used linear regression analyses with sex and the first 10 principal components as covariates. For linear regression, we used the bgenie tool[59]. We removed participants from the accelerometer discovery sample ($n = 91,112$), those who self-reported as being non-white British ($n = 68,428$), had abnormal genetic vs. self-reported sex mismatches ($n = 192$) or sex chromosome aneuploidy ($n = 652$). We also removed 48,658 participants due to relatedness. This left 278,374 UK Biobank participants who were not included in our discovery analysis, and other publicly available GWAS summary data from the MR-Base web platform[36]. Where possible, estimates were meta-analysed using a fixed effects model (inverse variance weighted average).

For analysis we retained index SNPs with $p < 5 \times 10^{-6}$ that were pruned for LD ($r^2 < 0.001$) and more than 10,000 kb apart. These loci ($p < 5 \times 10^{-6}$) explain more phenotypic variance than traditional genome-wide loci ($p < 5 \times 10^{-8}$) with negligible effects on horizontal pleiotropy (Supplementary Fig. 7). We then followed a number of steps to denote potential causality with disease outcomes (Supplementary Note 2). We used the maximum likelihood-based approach as our primary source of MR estimates. This is based on published simulation results suggesting that causal estimates obtained from summarised data using a likelihood-based model are almost as precise as those obtained from individual-level data[38]. Only likelihood-based risk estimates that were significant after Bonferroni-correction were considered. The potential effect of pleiotropy was evaluated by three complementary approaches, namely weighted median and weighted mode estimation[71,72], and the regression intercept from the MR-Egger method[73]. The sensitivity of causal inference to any individual genetic variant was tested by leave-one-out analysis. The Steiger test was used to provide evidence for the causal direction of the effect estimates[74]. Sensitivity analyses was also conducted to test whether instrument variables were associated with the following observational 'classic' confounders: income, smoking, area deprivation, and years of education[75]. Potential associations with confounders were subsequently followed-up by multivariate mendelian randomisation analyses[76] to investigate the robustness of

these associations to adjustment for potential confounders. Additionally, for MR associations that appeared robust to sensitivity and pleiotropy analyses, bi-directional MR was conducted to assess the direction of the causal estimate. Instrument variables for bi-directional analyses that relied on estimates generated from UK Biobank were re-estimated on participants not in our discovery GWAS accelerometer dataset. All MR analyses were conducted in R using the TwoSampleMR package[36].

**Investigating functional and biological mechanisms**. To investigate potential biological mechanisms underlying physical activity and sleep, we used DEPICT[42] at suggestive loci with $p < 1 \times 10^{-5}$ to identify: the most likely causal gene; reconstituted gene sets enriched for accelerometer-defined phenotype associations; and tissues and cell types in which genes from associated loci are highly expressed. Next, we used the FUMA web platform[28] to perform tissue enrichment analysis where the full distribution of SNPs was tested with 53 specific tissue types, based on GTEx[77] data. To identify significant enrichments, we accounted for multiple testing across 53 tissues and five traits ($p < 1.88 \times 10^{-4}$). To then identify pathways implicated by the activity and sleep associated loci, we used FUMA to perform hypergeometric tests on genes from these loci to investigate over-representations in genes predefined from the GWAS catalogue.

## Data availability

The summary phenotype variables that we have constructed will be made available as a part of the UK Biobank Returns Catalogue at http://biobank.ctsu.ox.ac.uk/crystal/docs.cgi?id = 1. All accelerometer data processing, feature extraction, and machine learning code are available at https://github.com/activityMonitoring. GWAS summary statistics, including with and without adjustment for BMI and sex as covariates, can be downloaded from https://doi.org/10.5287/bodleian:yJp6zZmdj.

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

## Acknowledgements

This research has been conducted using the UK Biobank resource under application number 11867. We would like to thank all participants for agreeing to volunteer in this research. Tugce Karaderi and Sven Hollowell contributed to data preparation. Matthew Willetts and Louis Aslett contributed to the development of machine-learning methods to identify activity phenotypes. We would also like to thank Benjamin Neale at the Broad Institute for advice on statistical testing for heritability differences between stratified groups. For data collection at Oxford and annotation we would like to acknowledge the support of Jonathan Gershuny, the UK Economic and Social Research Council (grant number ES/L011662/1), Charlie Foster, Paul Kelly, Teresa Harms, Emma Thomas, Karen Milton, Nicole Grey, Wong Tsz Yan, Salma Haque, Marina Topouzi, Sarah Darby, and Philipp Grunewald. The UK Biobank Activity Project and the collection of activity data from participants was funded by the Wellcome Trust (https://wellcome.ac.uk/) and the Medical Research Council (http://www.mrc.ac.uk/). The analysis was supported by: the National Institute for Health Research (NIHR) Oxford Biomedical Research Centre (BRC) [A.D., M.V.H., C.M.L.]; the British Heart Foundation Centre of Research Excellence at Oxford (http://www.cardioscience.ox.ac.uk/bhf-centre-of-research-excellence) [grant number RE/13/1/30181 to A.D.]; the Li Ka Shing Foundation (http://www.lksf.org/) [to A.D., C.M.L.]; Welcome Trust-SSI/John Fell funds Oxford [CML]; European Commission Horizon 2020 Widenlife [C.M.L.]; and NIH [grant number CRR00070 CR00.01 to C.M.L.]. M.V.H. works in a unit that receives funding from the UK Medical Research Council and is supported by a British Heart Foundation Intermediate Clinical Research Fellowship (FS/18/23/33512). We would also like to acknowledge the use of the University of Oxford Advanced Research Computing (ARC) facility in carrying out this work. http://dx.doi.org/10.5281/zenodo.22558 The MRC and Wellcome Trust played a key role in the decision to establish UK Biobank, and the accelerometer data collection. No funding bodies had any role in the analysis, decision to publish, or preparation of the manuscript.

## Author contributions

A.D., C.H., and C.M.L. conceived this study. A.D. developed and implemented the analyses, with input from T.F., and S.P. K.S.-B., and A.D. implemented the Mendelian Randomisation analysis with input from M.H. A.D. wrote the paper. A.D., C.M.L., S.P., M.H., C.H., T.F., and K.S.-B. were involved in interpreting results and editing the manuscript.

## Additional information

**Competing interests:** The authors declare no competing interests.

