## [Peer Review File · Nature Communications]

Reviewer #1 (Remarks to the Author):

This is a very well written and comprehensive analysis of data derived from a large sample, subsetted from the UKBiobank, who wore a multi day activity monitor. Analyses used a novel ML algorithm to extract estimates of overall activity, walking, sitting/standing and sleep and performed GWAS, heritability, MR and genetic/phenotypic correlations. The analysis revealed 10 novel loci that met stringent GWAS-significant threshold, 6 of which were novel. An analysis of a larger sample using self reported physical activity showed no significant findings, nicely demonstrating the importance of the phenotype in improving signal detection. MR showed that overall activity was causally related to lower fat and blood pressure. Bioinformatics data show that associations were for genes showing enrichment in the central nervous system and adrenal/pancreas.

There is a paucity of large scale genetic studies utilizing objective estimates for physical activity and sleep, and thus, this study provides an important contribution to the literature using data that are likely less biased and misclassified than self-report. The large sample size also allowed assessed of sex-specific associations. The confirmatory findings for some associations (e.g., MEIS1 for sleep) were reassuring and helps to identify associations that may be robust to moderate misclassification.

Overall, the approaches and data point to novel ways to use objective data from large numbers of individuals with techniques such as machine learning, and integrate these into genetic studies.

Despite this novelty and the many strengths of the study, there are areas that are somewhat ambiguous and analytical decisions that should be further provided and reconsidered to ensure the results were not influenced by confounders or model misspecification. There are also some issues in conceptualizing "movement" and sleep that are overly simplistic.

Conceptual overview: Physical activity and sleep are described as a "movement continuum". While the metrics analyzed are from movement sensors, sleep is a complex neurophysiological state distinct from movement. It would be important to clarify that while activity and sleep behaviors/activities are inter-related, these are distinct from "movement."

ML metrics and primary outcomes. While the authors note the advantages of objective data, "ground truth" for sleep was based on self-reported sleep diaries. Furthermore, the algorithm was based on a sample of only 57 individuals studied for one day (while the activity data was summarized over one week.) Suggest that the limitations of this approach, and possibility that "sleep" is influenced by subjective report, be included. The descriptives of the training sample as well as the overall analytical sample should be described- additional tables showing the

demographic and PA/sleep characteristics of each sample would help understand the comparability of these groups. Specifically, the distributions of PA and sleep measurements, for the analytical sample, by sex, would be helpful.

Sleep assessment: A focus of this study is not "sleep" but an estimate of "sleep duration"-this should be further clarified. As many studies have shown non-linear associations for sleep duration and even show that there are unique genetic associations for short vs long sleep, additional analyses should address linearity of assumptions and consider whether associations vary for short/long sleep. This study reports increasing sleep is associated with low education, cognition, etc--however, are these associations driven by the extreme long sleepers?

Confounders: Given the high lambdas, and consistent with most of the literature, why were population PCs not included in the analysis? For the overall analyses, sex should be included as a covariate, especially since activity monitors may vary in output by sex and BMI. Further analyses addressing whether associations change with adjustment for BMI (given that BMI could influence the measurement characteristics of the monitors) is needed. Adjusting for occupational category, SES, and overall health status may be helpful to improve interpretation of the findings as daily activities are clearly influenced by these factors. While reporting simple genetic correlations across traits is a common approach for exploring common genetic etiologies, there are many confounders that could explain such associations. Further adjusting for some key co-morbidities may help understand what correlations persist.

While sex differences in heritabilities were reported, was there evidence of a significant sex interaction?

Several of the figures had axes that were difficult to read.

Reviewer #2 (Remarks to the Author):

Comments to authors:

Summary:

Overall, this is a well written manuscript that presents a GWAS of physical activity and sleep derived from accelerometer data, along with a number of follow-up analyses using current approaches (BOLT-LMM heritability analyses, LD score regression, DEPICT, etc.). This represents a significant body of work and I believe work that has been carried out to a high standard. There are however several points of clarification regarding the methods that I think are required as well as the addition of some discussion around replication (as detailed in my comments below).

Major issues:

1. Results, para. 3 (p.5): You say “The remaining associations were known: two for ...” and “Nominal replication ...”. You should make clear in the text and in Table 1 which of these associations were discovered in the same dataset (i.e. in UK Biobank) – as far as I can tell this is all of them. I realise that this information is provided in that readers can look at the references themselves and also is more explicitly shown in Supplementary Table S3 but I think as it currently reads, the naïve reader might interpret this as an ‘independent’ replication of these loci which it is not. Indeed, it would be slightly concerning if the Klimentidis et al (2017) findings based on accelerometer phenotypes hadn’t been replicated.
2. Results, para. 5 (p.5): I would expect the estimates of phenotypic variance explained by the GWS loci to be biased upwards since this calculation has (necessarily) been done in the discovery cohort. I think this limitation should be explicitly recognised either here or in the discussion.
3. Results, para. 10 (p.7): You say the MR was conducted in “278,367 UK Biobank participants who were not included in our discovery analysis” could you be more explicit here that you are referring to the discovery cohort for the accelerometer analysis not the self-report physical activity discovery cohort? Also, I am slightly confused because in Table S9 you clearly state: “MR estimates were generated using outcome instruments estimated only in individuals that were not included in the current GWAS.” But then in Table S10, you say: “This includes data from all UK Biobank participants and other GWAS datasets available on MR-Base. SE and beta are in exposure SD units.” As you seem to acknowledge, including those individuals involved in the discovery analysis in the subsequent MR analysis would be inappropriate and could propagate any bias. This needs clarification (see further comment below).

4. Discussion (p.9-11): As far as I can see, nowhere in the discussion do you acknowledge the lack of independent replication as a limitation of this work – I believe that this is an oversight that should be addressed. I fully recognise the challenges associated with replicating such a study – presumably the UK Biobank cohort represents the largest (possibly by some margin) genotyped population with accelerometer data, and I also appreciate the fact that by using such a large cohort, some of the problems traditionally associated with using smaller discovery cohorts may not apply. However, I strongly feel that you should include a paragraph that discusses the lack of replication as a potential limitation and that in that paragraph you address which of the traditional problems you think do not apply here and for any remaining, how you have attempted to address them. It would also be interesting to see a power calculation done that explores what size cohort would be needed to replicate your findings and some indication as to whether such a cohort (with the necessary phenotypic data) exists (to the best of your knowledge) and relating to this, any intentions you have to replicate the work.

5. Methods, para.20 (p.19): As mentioned above (pt. 3), while you say the MR analysis was conducted using only those UK Biobank participants not in the accelerometer discovery sample the results presented in Table S10 suggest those individuals were used in some of the analyses. If this is the case, I would need convincing as to why this was the chosen approach, otherwise can you reword the header for Table S10.

6. Methods, para. 9 (p.15): You say: “Sensitivity analyses showed we did not need to include principal components ..” I think it is quite unusual not to fit PCs in UK Biobank analyses and that the potential for underlying substructure is quite high. Please could you elaborate on what these sensitivity analyses were. I would also like to see results presented (in supplementary) for GWS index SNPs (those presented in Table 1) using an alternative model with PCs fitted.

Minor issues:

7. Intro, para. 2 (p.3): Given the issue raised above regarding replication, I think it would be prudent to at this point bring to the reader’s attention the fact that the majority of current research in the area of genetic analyses (at least GWAS) of physical activity and sleep has been done using UK Biobank data. From what I can see, the references you currently provide in this paragraph (ref 8-12) all use UK Biobank as their primary dataset. Clearly this is because it is an excellent resource for this kind of analysis and thus it shouldn’t necessarily be badged as a disadvantage, but I think it is important to accurately set the scene at this point of the manuscript.

8. Results, para. 3 (p.5): You say: “Nominal replication ...”, although the criteria for nominal replication are stated in the methods, it would be nice to see it here also.

9. Results: Is there a point in the manuscript (at the end of GWAS result itself?) where you can say something like “From this point forward we only consider accelerometer traits” since as I understand it for most of the following analyses (e.g. genetic architecture, relationship with other traits) only the accelerometer phenotypes are considered. This makes sense as you didn’t find any GWS loci for the self-report measure(s) but still worth being explicit in this.

10. Results, para. 6 (p.6): You say: "Regions of the genome annotated ... functional categories and traits ($p < 2.38 \times 10^{-4}$, accounting for 21 categories)." Is it $p < 2.38 \times 10^{-4}$ because you're summarising across the traits, i.e. the biggest p-value was $p = 2.38 \times 10^{-4}$? Or is this the threshold for 'significance' (as suggested by 'accounting for 21 categories'). But in this case why doesn't it match the p-value in Fig. 2 ($p < 2.38 \times 10^{-4}$) or Fig S4 ($p < 4.76 \times 10^{-4}$) or the methods ($p < 3.57 \times 10^{-4}$). I am not bothered about the exact p-value used per se, but I do want to understand the process by which results have been firstly derived and then summarised across traits. Could you add some details to the methods (top of p.17) to address this issue?
11. Results, para. 10 (p.8): I cannot find the figures you quote for the MR result for body fat percentage in the supplementary tables (beta = -0.44, SE=0.047).
12. Results, para. 10 (p.8): Regarding the statement "directionality analyses found evidence in support of activity being proximal ..." – given that these kinds of analyses are relatively new in MR, could you provide more details on your interpretation of the results in Tables S11 & S12, in particular the Steiger test (what do these p-values mean?). This could be done here, in the Methods or in the Tables directly. Also, in the supplementary tables it suggests the Egger intercept tells you something about directionality but as you correctly state in the methods this tells you about the potential effect of pleiotropy.
13. Discussion, para. 2 (p.9): You say: "We note our current MR estimates are based on a limited number of SNPs.". It would be helpful if you provided some F-statistics in the results section.
14. Discussion, para. 3 (p.9): You say: "We observed an important role ... consistent with emerging literature." I think to use ref. 8 in this context is somewhat disingenuous since it is an analysis that, as I understand it, was based on essentially the same data.
15. Methods, para. 1 (p.12): You focus on the accelerometer cohort here but should also describe the wider cohort used in the self-reported physical activity analysis – did all UK Biobank participants fill in the relevant questionnaire for this part of the analysis, for example.
16. Methods, para. 13 (p.16) onwards: As mentioned above (pt. 9) – please be clear when you 'drop' the self-report traits.
17. Methods, para. 13 (p.17): As mentioned above (pt. 10) more detail is needed here to enable the reader to understand the process by which 'functional enrichment' results have been firstly derived and then summarised across traits (and therefore why there are several different p-value thresholds presented).
18. Methods, para.20 (p.19): As mentioned above (pt. 12), please can you provide more explanation regarding how you used the Steiger test.
19. Methods, para. 20 (p.19): As mentioned above (pt. 13), please can you report F-statistics to indicate the strength of the SNP-exposure relationship? I see you make some reference to this in row 2 of the S10 worksheet but it lacks any explanation/context so I not sure what this is referring to.

Typos, etc:

20. Results, para. 3 (p.4): In the sentence that starts “Our analysis identified 10 significant loci ...” you refer to Fig. S1 but it should be Fig. S2.

21. Typo in Table S9: “MR estimates were generated using outcome instruments estimated only in individuals that were not included in the current GWAS.”

Reviewer #3 (Remarks to the Author):

This a rich study identifying genetic variants contributing to individual differences in objectively assessed physical activity and sleep, which are regarded to be important health behaviors still amenable to meaningful intervention in the UKB participants’ age range. In over 90k UKB participants, SNP heritability for Axivity Ax3 wrist-worn triaxial accelerometer derived phenotypes was about 12% (higher in females than males) with 10 loci meeting GW significance levels. Demonstration through Mendelian Randomization of a potential causal effect of total daily physical activity on cardiometabolic risk factors is a strong reason to support publication of this work with subsequent availability of its summary statistics to the research community. Whether Nat Comm is the correct outlet rather than a specialist journal could be debated.

The paper is well-crafted and follows a clear structure with a GWA on five accelerometer phenotypes using stringent significance level and appropriate checks for stratification followed by

- Further lead-SNP conditional (GCTA), and gene-based (FUMA) association analysis.
- SNP-based heritability estimates for males and females (BOLT-LMM)
- Genetic correlation of sleep and activity with other GWAS’ed traits (LDreg)
- Enrichment of lower p-valued SNPs in conserved regions, tissue-specific gene expression, or specific biological pathways.
- Mendelian randomization (with various methods of pleiotropy control) to show a causal effect of physical activity on BP and body fat.

These analyses are generally performed state-of-the-art, although some of the reporting (in the supplements) could be improved (see below).

My general appreciation of this paper is balanced by a number of concerns related to phenotyping, (potential) selection bias, added value/novelty of the findings, and the somewhat limited progress in the understanding of the biology of the assessed behaviors.

Phenotyping:

The principle of using wearable-defined accelerometer phenotypes is a sound one. There are many technical complexities in separating movement from gravity or estimating energy expenditure from wrist (or other limbs) movement, but the authors are clearly at the forefront of accelerometry and I don't think a better job could have been done, taken the inherent complexity and our current knowledge. Notwithstanding, I am unsure whether the selection of the machine-learned phenotypes is optimal from a public health perspective. First of all, the mixture of sleep and physical activity is predicated entirely on the fact that they are derived from the same instrument by similar strategies. One can cleverly combine them in a single 24-hour (Canadian) guideline but the biological pathways leading from poor sleep to health and from low physical activity to health are still quite disparate. Their mixture in a single paper feels artificial. The mixture becomes frankly uncomfortable during enrichment analyses. What does the 'median value of the 5 physical activity traits and sleep' stand for in a biological sense? What possible use is, e.g. figure 2 (compared to the completely meaningful S4 on separate traits)?

Zooming in on the individual phenotypes I am even further worried by the choices made. Sleep duration is not the core health risk emerging from this vast literature; insomnia, sleep onset latency, sleep quality seem the more important traits. They are correlated with sleep duration but not perfectly and possibly not linearly. No sleep quality is detected by the accelerometer analysis, so not sure what to make of the two novel sleep loci. There are questions in UKB addressing sleep quality (perhaps one could explore the relation between objective, subjective duration and between both and measures of sleep quality/insomnia).

Total physical activity (TPA) and moderate-to-vigorous-intensity activity (MVPA) are meaningful and strategies of scoring these from a 7-day recording with the UKB device are sound. Novel loci SYT4 and ZNF423 are therefore interesting findings. However, it is MVPA that is the subject of PA guidelines (not total activity), so I don't see why TPA and not MVPA is favored in the analyses described in note S1 after establishing a high R_g between them.

The other phenotypes are somewhat curious. What does sitting/standing mean? There is a rapidly expanding literature on health risks associated with sitting, particularly TV viewing. However, this literature mentions standing as a potential countermeasure and there is a burgeoning industry of standing-at-work desks. I am not the one to claim a strong evidence-base for standing-as-a-cure. But the mixed bag of standing/sitting is not well-aligned conceptually with the sedentary behavior literature. Surely, a score of 80 hours standing/sitting a week is different for 80% standing and 20% sitting than for 20% standing and 80% sitting. I would now not know how to interpret the two novel loci for sitting/standing. The 'walking' phenotype is interesting (as is its 11% h^2), although evidence for health effects of walking are less solid than for MVPA. What I did not get is the extent of overlap between TPA and walking, or even between MVPA and (faster) walking. What happened to car driving? Was it/should it be made part of sitting? Finally, I assume that mixed activity and biking

were made part of MVPA, but this is also not clear. In short I am unsure of the overlap between the activity phenotypes and question the use of sitting/standing as a monolithic category.

Potential selection bias and their impact.

Out of 236519 invited participants, 132597 never responded. Were these people characterized by much more sedentary lifestyles? Put otherwise, are there in the rich UKB data set indications of a systematic bias in participation to 7 days of movement recording? A supplement describing either the absence of these biases or how detected biases could impact on the GWA would help the reader interpret the data.

Added value/novelty:

The paper you cite of Klimentidis et al. (<https://www.biorxiv.org/content/early/2017/11/07/179317>) also using accelerometer data from UKB has a substantial overlap with the work presented here. That paper is now well underway in its review process at a different journal (I am sorry for us living in such a small world). The accelerometer analyses for total and moderate-to-vigorous PA are clearly superior in the current paper but all loci (rs34517439, rs10851869, rs62055545, rs4747438) reported by Klimentidis et al. based on accelerometer are restated here (I would hesitate to call this replication- I suspect nearly the same data treatment was used). This impact negatively on the novelty value. On the flip side there are also worrisome differences that are left unexplained. Using much the same instruments, Klimentidis et al. did detect significant association with self-reported phenotypes, even at the same stringent $p < 10^{-9}$.

The cited paper by Jansen et al (<https://www.biorxiv.org/content/early/2018/01/30/214973>) on 1M subjects fully includes UKB and reports on quite a number of sleep variables. While sleep duration is better tagged by accelerometers, the overlap in knowledge (loci) between your paper with these previously reported results is substantial. This impact negatively on the novelty value.

Progress in the understanding of the biology of the assessed behaviors:

The post-GWAS analyses yields an important finding namely evolutionary conserved regions are enriched for PA SNPs. This means that animal models of the genetics of physical activity (cage movement, voluntary wheel running) might be quite informative for humans, something the authors may want to explicitly observe. Otherwise, however, a rather bland outcome is reported namely that genetic effects on sleep and physical activity 'work through the central nervous system/brain'. That can hardly come as a surprise for behavioral traits. Involvement of the adrenal gland/pancreas is certainly intriguing but no explanatory discussion or experimental follow-up is offered here. Neither is there a discussion on cerebellar involvement or frontal/ACC.

In depth discussion of FUMA outcomes is missing. Possibly rightfully so. But sentences like “physical activity traits share biology with multiple including intelligence, education, obesity, and cardiometabolic diseases” or “to cardiometabolic phenotypes, neurologic health” “showed genetic overlap with neurodegenerative diseases, mental health wellbeing, brain structure, function and connectivity” are uninformative and vague. They also reflect cherry picking because the majority of tested/expected genetic correlations (with CAD or cancer for example) were non-significant.

Although you invested quite a bit in the MR (directional) analyses on overweight and blood pressure – the discussion does not present clear summary statements of the findings and what they mean. It is all left a bit in the air.

In the abstract you rightfully express concerns that “the etiology of PA and sleep is poorly understood”. How has your work changed this situation? It would be good to make that more visible and explicit. At present I cannot place the remark in the introduction that “the associated loci reveal pathways in the central nervous system that help advance our understanding of biological processes underlying human sleep and activity behaviours”.

MINOR:

-Affiliation 11 is erroneously written as 10.

-In the abstract, giving the male and female h^2 estimates would be more informative than the difference ; p-values can be maintained.

-Page 4, figure S1 should probably be figure 1

-How many men and women participated? What does the sex difference in SNP- h^2 mean? Larger effects for the same genes in females? If so by which mechanism? For $R_g = 0.89$ in MVPA - was this significantly different from 1?

-Table S6-S7 are very difficult to read as they are not organized by accelerometer trait (like table S3). I suggest you switch columns 1 and 3 and sort by accelerometer trait.

- Obviously we cannot conclude from tables S6-S9 that all other traits tested (832 - 158 + 4000 – 115 + N_NHGRI - 5) but not found significant, therefore do not share a genetic basis with physical activity or sleep duration. I found this paragraph to generate a lot of concern – the positive correlations can reflect many different causal, mediational or pleiotropic scenario's and the negative ones would need to be qualified by power considerations. But if it must be, I would have preferred the use of MVPA over TPA.

- Figure S6 on SNP rs56194509 is unreadable, even viewed at high resolution at my screen. Also, unclear what triangles and color coding mean to convey.

- Important details on the MR analyses are missing. What is the complete list of traits selected for the MR pipeline in Note S2 – is that what is seen in table S10? Which SNPs were retained as the genetic instrument for the various accelerometer traits?

- Why is there a difference between results in S9 and S10 for e.g. body fat and overall activity? The order of S10 – S9 – S11 – S12ab would have made more sense, I think.

- Table S10 generates the same question on negative outcomes – surely the absence of MR is not proof of absence of causality, or is it? (Again, I would have preferred the use of MVPA over TPA).

- Figure S7 : text printed through the bars is hard to read.

- Methods: how many SNPs were used in genetic association for self-reported PA with the 351,154 participants?

- Thank you for your willingness to share the summary phenotype variables and your scoring algorithms.

Comments to address (response in red)

We would like to thank the reviewers for their positive and constructive comments on this manuscript. Based on this feedback, we conducted a substantial re-analysis that has resulted in a further improved manuscript. We now take into account a more cautious and comprehensive consideration of the accelerometer-defined phenotypes and also acknowledge the lack of replication in the discovery GWAS. Here are the major changes we have made:

- Accelerometer phenotypes have been redefined to identify: overall activity (unchanged), sleep duration (rather than “sleep”), sedentary behaviour (rather than “sit/stand”), walking (unchanged), and moderate intensity activity (which now does not overlap with the other behavioural traits)
- Training of machine-learned phenotypes now takes place on a larger training dataset of 153 participants (versus 57 previously).
- GWAS discovery now includes SNPs from the recently updated UK Biobank imputation dataset that includes UK10k+ imputed variants. In practice few of these additional variants passed quality control minor allele frequency and imputation quality score criteria.
- After much double-checking, we realise a mistake was made in the preparation of the self-reported overall activity phenotype. Rerunning this analysis correctly results in 11 genome wide significant loci. Therefore, one of our initial conclusions on the futility of self-report measures of activity was clearly wrong - both measures are helpful and complementary. Given our original paper was already information dense, we now just report results for the accelerometer phenotypes, and discuss the pros and cons of objective- versus subjective- defined behavioural phenotypes in the discussion section.
- All Mendelian Randomisation analyses now ensure participants were not in the accelerometer discovery GWAS dataset. We have also increased the number of MR instrument variables after noting no effects with respect to horizontal pleiotropy after selecting instrument variables with a less stringent p-value (5×10^{-6}) from the discovery GWAS (see figure S5).

Please find our responses to specific comments from the reviewers below, plus details of how we responded to them.

Response to reviewer #1 comments:

Conceptual overview: Physical activity and sleep are described as a "movement continuum". While the metrics analyzed are from movement sensors, sleep is a complex neurophysiological state distinct from movement. It would be important to clarify that while activity and sleep behaviors/activities are inter-related, these are distinct from "movement."

Authors' response: We agree with this comment, and have now updated the manuscript to reflect that we are focused on "sleep duration" and "physical activity behaviours", without making any claims that they share a continuum.

Changes to the paper: Introductory paragraph is now updated, and all mentions of "sleep" phenotype are updated to the more explicitly defined "sleep duration" phenotype.

ML metrics and primary outcomes. While the authors note the advantages of objective data, "ground truth" for sleep was based on self-reported sleep diaries. Furthermore, the algorithm was based on a sample of only 57 individuals studied for one day (while the activity data was summarized over one week.) Suggest that the limitations of this approach, and possibility that "sleep" is influenced by subjective report, be included. The descriptives of the training sample as well as the overall analytical sample should be described- additional tables showing the demographic and PA/sleep characteristics of each sample would help understand the comparability of these groups. Specifically, the distributions of PA and sleep measurements, for the analytical sample, by sex, would be helpful.

Authors' response: At the time of submission, we had only annotated groundtruth data for 57 participants. Since then, we hired and trained more assistants; using the same protocol as before, to complete annotation of all available ground-truth image data (now on 153 participants). We have reported the age and sex composition of this training sample, but not the PA/sleep characteristics due to the one-versus seven- day wear-times between both datasets. We believe this is not a substantial issue as the size of our training dataset has helped provide a diverse set of representative behaviours, rather than individuals, which is important for model development.

Changes to the paper: Results paragraph updated to state *"We trained and evaluated the model in 153 free-living individuals (mean age=42, female n=100) to distinguish between activity states, evaluated against a wearable camera and time-use diary ground-truth"*. Discussion section updated to state *"While the participants in our machine learning free-living training dataset are not a random subsample of the UK Biobank study, the size of our training dataset has helped provide a diverse set of representative behaviours, rather than individuals, which is important for model development"*.

Sleep assessment: As many studies have shown non-linear associations for sleep duration and even show that there are unique genetic associations for short vs long sleep, additional analyses should address linearity of assumptions and consider whether associations vary for

short/long sleep. This study reports increasing sleep in associated with low education, cognition, etc--however, are these associations driven by the extreme long sleepers?

Authors' response: In preparation for this revised manuscript we ran additional GWAS on short and long sleep, defined as those participants in the bottom or top quintiles respectively. We found three loci for short sleep, and one locus for long sleep, all of which were already identified for sleep duration. Thus it appears those who are extreme short or long sleepers do not drive our current results. However, we agree with the reviewer that it will be important for the field to conduct further investigations into the linearity of sleep assumptions. For example, in the discussion section we recognise future work will be also required for MR sleep analysis to account for the likely U-shape relationship between sleep and disease outcomes, rather than a linear relationship as assumed in current models.

Changes to the paper: New paragraph in results section that states *“To account for the likely non-linear U-shape relationship between sleep and disease outcomes⁴⁹, we ran additional GWAS on short and long sleep, defined as those in the bottom or top quintiles for sleep duration respectively. We found three loci for short sleep (near PAX8-AS1, MEIS1, and MAPK8IP1) that were already identified for sleep duration. We found one locus for long sleep duration (near PAX8-AS1) that was also already identified for sleep duration. This indicates that our sleep duration results are not driven by individuals who are very short or long sleepers.”*

Confounders: Given the high lambdas, and consistent with most of the literature, why were populations PCs not included in the analysis?

Authors' response: Linear-mixed models can correct for population and relatedness structure in the data. However, as mentioned in the methods text we conducted a sensitivity analysis that showed we did not need to include principal components of ancestry as covariates in our linear mixed models. The text has now been updated to report the specific result for this sensitivity analysis.

Changes to the paper: Methods text updated to state *“Sensitivity analysis on the overall activity trait showed we did not need to include principal components of ancestry as covariates in our linear mixed models ($\lambda=1.20$ v 1.20 , LD intercept = 1.0085 vs 1.0092 , $R_g = 0.99$).”*

Confounders: For the overall analyses, sex should be included as a covariate, especially since activity monitors may vary in output by sex and BMI. Further analyses addressing whether associations change with adjustment for BMI (given that BMI could influence the measurement characteristics of the monitors) is needed. Adjusting for occupational category, SES, and overall health status may be helpful to improve interpretation of the findings as daily activities are clearly influenced by these factors. While reporting simple genetic correlations across traits is a common approach for exploring common genetic etiologies, there are many confounders that could explain such associations. Further adjusting for some key co-morbidities may help understand what correlations persist.

Authors' response: We gave this issue much thought before submitting our original manuscript. However we decided against including a range of covariates to avoid the introduction of collider bias that would have an effect on all downstream analyses. We do recognise that this is an area of scientific uncertainty where some papers show no effect of collider bias (*Holmes & Davey Smith; Mol Psychiatry 2018 doi:10.1038/s41380-018-0037-1*), whereas others show this to be an issue with waist-hip-ratio adjusted for body mass index (BMI) was inversely associated with BMI being positively associated with WHR (*Emdin et al.; JAMA 2017;317(6):626-634*).

To further satisfy ourselves that the introduction of common covariates is not necessary we conducted three additional GWAS on the overall activity trait, using different covariates. We found the inclusion of sex and BMI were strongly correlated with our main discovery GWAS ($R_G=0.93-0.99$), and thus had minimal impact.

Changes to the paper: We added season-of-wear as an additional covariate, which we believe is unlikely to increase risk of collider bias in downstream analysis. Two sentences have been added to methods: *“We decided against including too many covariates to avoid the introduction of any possible collider bias that could have an effect on downstream analyses. In addition, we found little difference with the inclusion of: 1) sex as an additional covariate ($R_G=0.99$); 2) BMI as an additional covariate ($R_G=0.93$); and 3) both sex and BMI as additional covariates ($R_G=0.93$).”*

While sex differences in heritabilities were reported, was there evidence of a significant sex interaction?

Authors' response: We did not find evidence of sexual dimorphism for heterogeneity of effects between men and women ($p<5\times 10^{-9}$).

Changes to the paper: None (the above statement is in the *“Sexual dimorphism in physical activity and sleep” results subsection*).

Several of the figures had axes that were difficult to read.

Authors' response: We have updated the axis labels for Figures S3 and S7. We will of course be very happy to work with the production team to further amend figure fonts if necessary.

Changes to the paper: Axes updated in supplement figures S3 and S7. Original figures 2 & 3 are now removed.

Response to reviewer #2 comments:

Comments to authors:

Summary:

Major issues:

1. Results, para. 3 (p.5): You say “The remaining associations were known: two for ...” and “Nominal replication ...”. You should make clear in the text and in Table 1 which of these associations were discovered in the same dataset (i.e. in UK Biobank) – as far as I can tell this is all of them. I realise that this information is provided in that readers can look at the references themselves and also is more explicitly shown in Supplementary Table S3 but I think as it currently reads, the naïve reader might interpret this as an ‘independent’ replication of these loci which it is not. Indeed, it would be slightly concerning if the Klimentidis et al (2017) findings based on accelerometer phenotypes hadn’t been replicated.

Authors’ response: We agree and have now updated the text to more clearly reflect that the associations in question are not ‘replicated’ and instead have also been reported by others using the same UK Biobank dataset.

Changes to the paper: Results text updated to reflect “*We found concordance for loci previously reported using UK Biobank data for <traits>...*”. Supplement table 3 updated with new title “*Table S3 | Results in this UK Biobank accelerometer cohort for genetic variants previously reported to be associated with traits related to physical activity and sleep accelerometer measures in UK Biobank and other datasets. Significant associations ($p < 2.3 \times 10^{-4}$, accounting for 213 candidate loci) are highlighted in green.*”

2. Results, para. 5 (p.5): I would expect the estimates of phenotypic variance explained by the GWS loci to be biased upwards since this calculation has (necessarily) been done in the discovery cohort. I think this limitation should be explicitly recognised either here or in the discussion.

Authors’ response: We agree and have now updated the discussion section text.

Changes to the paper: Discussion section now contains an extra sentence stating: “*Replication in other cohorts would also help refine estimates of phenotypic variance explained by genome wide significant loci, which are currently calculated in the discovery dataset reported in this paper.*”

3. Results, para. 10 (p.7): You say the MR was conducted in “278,367 UK Biobank participants who were not included in our discovery analysis” could you be more explicit here that you are referring to the discovery cohort for the accelerometer analysis not the self-report physical activity discovery cohort? Also, I am slightly confused because in Table S9 you clearly state: “MR estimates were generated using outcome instruments estimated only in individuals that were not included in the current GWAS.” But then in Table S10, you say: “This includes data from all UK Biobank participants and other GWAS datasets available on

MR-Base. SE and beta are in exposure SD units.” As you seem to acknowledge, including those individuals involved in the discovery analysis in the subsequent MR analysis would be inappropriate and could propagate any bias. This needs clarification (see further comment below).

&

5. Methods, para.20 (p.19): As mentioned above (pt. 3), while you say the MR analysis was conducted using only those UK Biobank participants not in the accelerometer discovery sample the results presented in Table S10 suggest those individuals were used in some of the analyses. If this is the case, I would need convincing as to why this was the chosen approach, otherwise can you reword the header for Table S10.

Authors’ response: We agree that our original table was confusing and unnecessary. Therefore, we have now updated all Mendelian Randomisation supplement tables to ensure participants were not in the accelerometer discovery dataset.

Changes to the paper: Supplement tables S10-S13 now only report results for analyses where participants were not in the accelerometer discovery dataset.

4. Discussion (p.9-11): As far as I can see, nowhere in the discussion do you acknowledge the lack of independent replication as a limitation of this work – I believe that this is an oversight that should be addressed. I fully recognise the challenges associated with replicating such a study – presumably the UK Biobank cohort represents the largest (possibly by some margin) genotyped population with accelerometer data, and I also appreciate the fact that by using such a large cohort, some of the problems traditionally associated with using smaller discovery cohorts may not apply. However, I strongly feel that you should include a paragraph that discusses the lack of replication as a potential limitation and that in that paragraph you address which of the traditional problems you think do not apply here and for any remaining, how you have attempted to address them. It would also be interesting to see a power calculation done that explores what size cohort would be needed to replicate your findings and some indication as to whether such a cohort (with the necessary phenotypic data) exists (to the best of your knowledge) and relating to this, any intentions you have to replicate the work.

Authors’ response: We agree and have now updated the discussion text to place a stronger emphasis on the need for replication in future studies.

Changes to the paper: Relevant discussion section paragraph updated to now include following text: *“Given the cohort age range (45-80 years) and inclusion of European ancestry individuals only, we cannot necessarily assume our results generalise to other age groups or ancestral populations. Therefore, replication of our findings in non-European populations with both genetic and accelerometer-measures of sleep duration and physical activity will be important in future. Replication in other cohorts would also help refine estimates of phenotypic variance explained by genome wide significant loci, which are currently calculated in the discovery dataset reported in this paper.”*

6. Methods, para. 9 (p.15): You say: “Sensitivity analyses showed we did not need to include principal components ..” I think it is quite unusual not to fit PCs in UK Biobank analyses and that the potential for underlying substructure is quite high. Please could you elaborate on what these sensitivity analyses were. I would also like to see results presented (in supplementary) for GWS index SNPs (those presented in Table 1) using an alternative model with PCs fitted.

Authors’ response: Linear-mixed models can correct for population and relatedness structure in the data. However, as mentioned in the methods text we conducted a sensitivity analysis that showed we did not need to include principal components of ancestry as covariates in our linear mixed models. The text has now been updated to report the specific result for this sensitivity analysis.

Changes to the paper: Methods text updated to state “*Sensitivity analysis on the overall activity trait showed we did not need to include principal components of ancestry as covariates in our linear mixed models ($\lambda=1.20$ v 1.20 , LD intercept = 1.0085 vs 1.0092 , $R_g = 0.99$).*”

Minor issues:

7. Intro, para. 2 (p.3): Given the issue raised above regarding replication, I think it would be prudent to at this point bring to the reader’s attention the fact that the majority of current research in the area of genetic analyses (at least GWAS) of physical activity and sleep has been done using UK Biobank data. From what I can see, the references you currently provide in this paragraph (ref 8-12) all use UK Biobank as their primary dataset. Clearly this is because it is an excellent resource for this kind of analysis and thus it shouldn’t necessarily be badged as a disadvantage, but I think it is important to accurately set the scene at this point of the manuscript.

Authors’ response: We agree with this point and have updated the text accordingly.

Changes to the paper: Introduction section sentence updated to “*Recent GWAS have also reported common variants that are associated with sleep duration in UK Biobank^{10–12}.*”

8. Results, para. 3 (p.5): You say: “Nominal replication ...”, although the criteria for nominal replication are stated in the methods, it would be nice to see it here also.

Authors’ response: We feel that this section is already quite information dense, so have decided to leave the criteria in the methods and supplement table for the interested reader.

Changes to the paper: Relevant sentence updated to “*We found concordance for loci previously reported using UK Biobank data for <traits>*”.

9. Results: Is there a point in the manuscript (at the end of GWAS result itself?) where you can say something like “From this point forward we only consider accelerometer traits” since as I understand it for most of the following analyses (e.g. genetic architecture, relationship with other traits) only the accelerometer phenotypes are considered. This makes sense as you didn’t find any GWS loci for the self-report measure(s) but still worth being explicit in this.

&

16. Methods, para. 13 (p.16) onwards: As mentioned above (pt. 9) – please be clear when you ‘drop’ the self-report traits.

&

15. Methods, para. 1 (p.12): You focus on the accelerometer cohort here but should also describe the wider cohort used in the self-reported physical activity analysis – did all UK Biobank participants fill in the relevant questionnaire for this part of the analysis, for example.

Authors’ response: As mentioned at the very start, we now realise a mistake was made in the preparation of the self-reported overall activity phenotype. Given our original paper was already information dense, we now just report results for the accelerometer phenotypes, and discuss the pros and cons of objective- versus subjective- defined behavioural phenotypes in the discussion section.

Changes to the paper: Removal of analyses comparing self-reported and objective measures of physical activity.

10. Results, para. 6 (p.6): You say: “Regions of the genome annotated ... functional categories and traits ($p < 2.38 \times 10^{-4}$, accounting for 21 categories).” Is it $p < 2.38 \times 10^{-4}$ because you’re summarising across the traits, i.e. the biggest p-value was $p = 2.38 \times 10^{-4}$? Or is this the threshold for ‘significance’ (as suggested by ‘accounting for 21 categories’). But in this case why doesn’t it match the p-value in Fig. 2 ($p < 2.38 \times 10^{-4}$) or Fig S4 ($p < 4.76 \times 10^{-4}$) or the methods ($p < 3.57 \times 10^{-4}$). I am not bothered about the exact p-value used per se, but I do want to understand the process by which results have been firstly derived and then summarised across traits. Could you add some details to the methods (top of p.17) to address this issue?

&

17. Methods, para. 13 (p.17): As mentioned above (pt. 10) more detail is needed here to enable the reader to understand the process by which ‘functional enrichment’ results have been firstly derived and then summarised across traits (and therefore why there are several different p-value thresholds presented).

Authors’ response: We thank the reviewer for pointing out this inconsistency.

Changes to the paper: Results text now more is consistent with figure captions, including text to explicitly state thresholds for 1) tissue types and 2) functional categories: “*We identified significant tissue enrichments ($p < 1 \times 10^{-3}$, accounting for 10 tissue types and five traits) ... Regions of the genome annotated as conserved across mammals were enriched ($p < 4.7 \times 10^{-4}$, accounting for 21 functional categories and five traits)*”

11. Results, para. 10 (p.8): I cannot find the figures you quote for the MR result for body fat percentage in the supplementary tables (beta = -0.44, SE=0.047).

Authors' response: These can be found in Table S10A.

Changes to the paper: Caption for Table S10A updated to: *“Meta analysis of Mendelian Randomization results for adiposity traits using GIANT and UK Biobank participants not in accelerometer discovery dataset. SE and beta are in exposure SD units.”*

12. Results, para. 10 (p.8): Regarding the statement “directionality analyses found evidence in support of activity being proximal ...” – given that these kinds of analyses are relatively new in MR, could you provide more details on your interpretation of the results in Tables S11 & S12, in particular the Steiger test (what do these p-values mean?). This could be done here, in the Methods or in the Tables directly. Also, in the supplementary tables it suggests the Egger intercept tells you something about directionality but as you correctly state in the methods this tells you about the potential effect of pleiotropy.

&

18. Methods, para.20 (p.19): As mentioned above (pt. 12), please can you provide more explanation regarding how you used the Steiger test.

Authors' response: We agree that more details on interpretation of the Mendelian Randomization results should be added. To ease readability we have added these interpretations to each of the supplement tables.

Changes to the paper: Interpretation section added to Tables S10-13B. e.g. for Table S10 this note is: *“Interpretation: As per Supplement Note 2, we firstly seek maximum likelihood estimate p-values that pass the multiple testing threshold (highlighted in green below). For overall activity, we found preliminary indications for: body mass index, body fat %, Alzheimer's, systolic & diastolic blood pressure, and diagnosis of hypertension. For sleep duration we found preliminary indications for: body fat % and hypertension diagnosis. The overall activity and sleep duration estimates are in a similar direction when using Median, Modal and MR-Egger estimates too.”*

13. Discussion, para. 2 (p.9): You say: “We note our current MR estimates are based on a limited number of SNPs.”. It would be helpful if you provided some F-statistics in the results section.

&

19. Methods, para. 20 (p.19): As mentioned above (pt. 13), please can you report F-statistics to indicate the strength of the SNP-exposure relationship? I see you make some reference to this in row 2 of the S10 worksheet but it lacks any explanation/context so I not sure what this is referring to.

Authors' response: We now select instrument variables at loci with $p < 5 \times 10^{-6}$, as they explain more phenotypic variance than traditional genome wide loci ($p < 5 \times 10^{-8}$) with negligible effects on horizontal pleiotropy. However, we have also explicitly added F-statistics to Table S11.

Changes to the paper: We have now added F-statistics values for each choice of instrument variable to “Table S11 | Effect of instrument variable selection by GWAS discovery threshold on Mendelian Randomization analysis outcomes. Beta and SE are in exposure SD units.” With an interpretation of “*The strength of instruments is acceptable with an average F-statistic score of 24 (range 10 - 48).*”

14. Discussion, para. 3 (p.9): You say: “We observed an important role ... consistent with emerging literature.” I think to use ref. 8 in this context is somewhat disingenuous since it is an analysis that, as I understand it, was based on essentially the same data.

Authors’ response: We agree that this text could be improved, and have updated the manuscript accordingly.

Changes to the paper: Relevant paragraph in discussion is now updated to “*We observed an important role for the central nervous system with respect to physical activity and sleep, consistent with other studies that have used UK Biobank data⁸*”.

Typos, etc:

&

20. Results, para. 3 (p.4): In the sentence that starts “Our analysis identified 10 significant loci ...” you refer to Fig. S1 but it should be Fig. S2.

&

21. Typo in Table S9: “MR estimates were generated using outcome instruments estimated only in individuals that were not included in the current GWAS.”

Authors’ response: Thank you for pointing this out, we have corrected the relevant text.

Changes to the paper: Both typos are now corrected.

Response to reviewer #3 comments:

Phenotyping:

The mixture of sleep and physical activity is predicated entirely on the fact that they are derived from the same instrument by similar strategies. One can cleverly combine them in a single 24-hour (Canadian) guideline but the biological pathways leading from poor sleep to health and from low physical activity to health are still quite disparate. Their mixture in a single paper feels artificial. The mixture becomes frankly uncomfortable during enrichment analyses. What does the ‘median value of the 5 physical activity traits and sleep’ stand for in a biological sense? What possible use is, e.g. figure 2 (compared to the completely meaningful S4 on separate traits)?

Authors’ response: We agree with this comment, and have now updated the manuscript to reflect that we are focused on “sleep duration” and “physical activity behaviours”, without making any claims that they share a continuum.

Changes to the paper: Introductory paragraph is now updated, and all mentions of “sleep” phenotype are updated to the more explicitly defined “sleep duration” phenotype. We have also removed the original enrichment plots of ‘median’ values across the traits.

Zooming in on the individual phenotypes I am even further worried by the choices made. Sleep duration is not the core health risk emerging from this vast literature; insomnia, sleep onset latency, sleep quality seem the more important traits. They are correlated with sleep duration but not perfectly and possibly not linearly. No sleep quality is detected by the accelerometer analysis, so not sure what to make of the two novel sleep loci. There are questions in UKB addressing sleep quality (perhaps one could explore the relation between objective, subjective duration and between both and measures of sleep quality/insomnia).

Authors’ response: We agree that the UK Biobank dataset creates an exciting opportunity to discover more about the genetic architecture and biology of sleep. For example, Mike Weedon’s team at Exeter have recently posted two papers on the bioRxiv on self-reported and accelerometer measures of dominant sleep periods, and a detailed analysis of their subtypes - we are supportive of their work. Our machine-learned phenotypes include a complementary single measure of sleep duration, which we compare to the Exeter team (and cite their work). However, rather than repeating their analysis on further sleep traits, we have instead focused on the genetics of other machine-learned activity behaviours in this work (sedentary, walking, moderate-intensity and overall activity).

Changes to the paper: Rather than duplicating the work of others, we retain our focus on sleep duration and physical activity behaviours.

Total physical activity (TPA) and moderate-to-vigorous-intensity activity (MVPA) are meaningful and strategies of scoring these from a 7-day recording with the UKB device are sound. Novel loci SYT4 and ZNF423 are therefore interesting findings. However, it is MVPA that is the subject of PA guidelines (not total activity), so I don’t see why TPA and

not MVPA is favored in the analyses described in note S1 after establishing a high Rg between them.

Authors' response: While current physical activity guidelines do focus on moderate intensity activity, this is due to their reliance on self-report measures which were typically designed to only capture moderate intensity activity. This will likely change as future guidelines include objective measures of activity can assess the utility of other forms of activity (such as overall activity, light-intensity activity, and the behaviours proposed in this manuscript). A problem with using MVPA versus TPA is the reliance of selecting an appropriate y-axis threshold for defining whether one is in moderate intensity activity or not. The most commonly such used threshold is set at 100mg, but is based on a study of 30 Norwegian adults with a mean age of 34 (Hildebrand 2014, Med Sci Sport Exerc). The TPA measure does not require the introduction of thresholds and was therefore decided as the recommended summary measure of use for UK Biobank participants. For the same reason, we selected the total physical activity metric as a point of focus in this manuscript.

Changes to the paper: We now report results for moderate activity, which no longer overlaps with the other behavioural phenotypes as it is now defined via machine learning of movement patterns in 153 UK adults (versus 30 Norwegian adults in previous threshold-reliant approach).

The other phenotypes are somewhat curious. What does sitting/standing mean? There is a rapidly expanding literature on health risks associated with sitting, particularly TV viewing. However, this literature mentions standing as a potential countermeasure and there is a burgeoning industry of standing-at-work desks. I am not the one to claim a strong evidence-base for standing-as-a-cure. But the mixed bag of standing/sitting is not well-aligned conceptually with the sedentary behavior literature. Surely, a score of 80 hours standing/sitting a week is different for 80% standing and 20% sitting than for 20% standing and 80% sitting. I would now not know how to interpret the two novel loci for sitting/standing.

Authors' response: This is an excellent point and we have now retrained our model to identify a sedentary behaviour phenotype, rather than a sitting phenotype. The sedentary phenotype follows the definition proposed by an international Sedentary Behavior Research Network (SBRN) Terminology Consensus Project (Tremblay 2017, Intl J Behv Nutr Phys Act).

Changes to the paper: All results now updated and methods text now states “*sedentary behaviour is one which has a MET energy expenditure score of ≤ 1.5 and occurs in a sitting, lying, or reclining posture*⁶⁰. As an exception to this rule we also categorised the following annotations as sedentary behaviour: driving (code 16010) where assigned MET score is 2.5, and some instances of non-desk work (code 21010 rather than 5080). The complete list of image derived annotations and the phenotype class they were mapped to are available in Table S1.”

The ‘walking’ phenotype is interesting (as is its 11% h2), although evidence for health effects of walking are less solid than for MVPA. What I did not get is the extent of overlap between TPA and walking, or even between MVPA and (faster) walking. What happened to car driving? Was it/should it be made part of sitting? Finally, I assume that mixed activity and biking were made part of MVPA, but this is also not clear. In short I am unsure of the overlap between the activity phenotypes and question the use of sitting/standing as a monolithic category.

Authors’ response: On further reflection we also didn’t like that there was overlap between MVPA (threshold-based approach) and the other behaviour traits (all machine-learned). We therefore retrained our model to identify moderate activities (examples include running, playing tennis, shoveling mud, etc.) listed in new Table S1. This machine-learned moderate activity phenotype is complementary to the other behavioural phenotypes and is likely more robust (trained on 153 UK adults versus 30 Norwegian adults in previous approach).

Changes to the paper: All results throughout manuscript are updated to reflect moderate activity no longer overlaps other behavioural traits.

Potential selection bias and their impact. Out of 236519 invited participants, 132597 never responded. Were these people characterized by much more sedentary lifestyles? Put otherwise, are there in the rich UKB data set indications of a systematic bias in participation to 7 days of movement recording? A supplement describing either the absence of these biases or how detected biases could impact on the GWA would help the reader interpret the data.

Authors’ response: There were no significant differences in consent by self-reported physical activity status, and we have now updated the text to reflect this.

Changes to the paper: Method section text describing our collection of accelerometer data in UK Biobank has been updated to also state: “*Participants who were slightly more likely to consent included: women, those aged 55-74 years, those with higher socio-economic status, better physical health status, and less time since the baseline assessment (all odds ratios <1.1). There were no significant differences in consent by self-reported physical activity status.*”

Added value/novelty: The paper you cite of Klimentidis et al. (<https://www.biorxiv.org/content/early/2017/11/07/179317>) also using accelerometer data from UKB has a substantial overlap with the work presented here. That paper is now well underway in its review process at a different journal (I am sorry for us living in such a small world). The accelerometer analyses for total and moderate-to-vigorous PA are clearly superior in the current paper but all loci (rs34517439, rs10851869, rs62055545, rs4747438) reported by Klimentidis et al. based on accelerometer are restated here (I would hesitate to call this replication- I suspect nearly the same data treatment was used). This impact negatively on the novelty value.

Authors’ response: We were pleased to see the Klimentidis bioRxiv paper as it made use of the summary phenotypes we returned to the UK Biobank resource. We feel our manuscript

provides due credit to any loci first reported in the Klimentidis paper. However, we note that there are substantial differences between manuscripts, where our work includes: the derivation of novel machine-learned phenotypes; GWAS discovery in a larger sample of participants (using linear mixed models versus traditional linear regression models); and Mendelian Randomization analyses.

Changes to the paper: None in relation to this comment for the reasons just stated.

On the flip side there are also worrisome differences that are left unexplained. Using much the same instruments, Klimentidis et al. did detect significant association with self-reported phenotypes, even at the same stringent $p < 10^{-9}$.

Authors' response: As mentioned at the very start, we now realise a mistake was made in the preparation of the self-reported overall activity phenotype. Given our original paper was already information dense, we now just report results for the accelerometer phenotypes, and discuss the pros and cons of objective- versus subjective- defined behavioural phenotypes in the discussion section, with reference to the Klimentidis self-report findings.

Changes to the paper: Removal of analyses comparing self-reported versus physical activity information sources. Discussion section paragraph has been updated, to *“Our work demonstrates the utility of large-scale objective measures to identify genetic associations with activity phenotypes. For example, we found that previous self-reported literature has significantly underestimated heritability for physical activity⁸ (~5% vs ~20% for accelerometer-measured traits). The larger-sample sizes that are available for self-reported phenotypes results in more identified loci⁸, however heritability and potential explained variance appears higher for the objective traits ... It is likely that combination of self-report and objective measures will be needed to further our understanding of the genes and biological pathways that underpin physical activity.”*

The cited paper by Jansen et al (<https://www.biorxiv.org/content/early/2018/01/30/214973>) on 1M subjects fully includes UKB and reports on quite a number of sleep variables. While sleep duration is better tagged by accelerometers, the overlap in knowledge (loci) between your paper with these previously reported results is substantial. This impact negatively on the novelty value.

Authors' response: As mentioned earlier, there are a number of papers coming out soon on the genetic architecture and biology of sleep (the Jansen paper, Exeter papers, and Harvard papers). Our machine-learned phenotypes include a complementary single measure of sleep duration, which we have compared to these other bioRxiv papers (and cite their work). However, rather than repeating their analyses on further sleep traits, we have instead focused on the genetics of other machine-learned activity behaviours in this work (sedentary, walking, moderate-intensity and overall activity).

Changes to the paper: We have cited papers from the Exeter and Harvard groups that have been posted to the bioRxiv since this manuscript was originally submitted in early February. These are updated in the text and table S4.

Progress in the understanding of the biology of the assessed behaviors: The post-GWAS analyses yields an important finding namely evolutionary conserved regions are enriched for PA SNPs. This means that animal models of the genetics of physical activity (cage movement, voluntary wheel running) might be quite informative for humans, something the authors may want to explicitly observe.

Authors' response: We agree

Changes to the paper: Discussion text updated with new sentence stating: *“Animal models might be informative to answer such questions, as our heritability partitioning analysis supports a key role for physical activity and sleep behaviors throughout mammalian evolution.”*

Otherwise, however, a rather bland outcome is reported namely that genetic effects on sleep and physical activity ‘work through the central nervous system/brain’. That can hardly come as a surprise for behavioral traits. Involvement of the adrenal gland/pancreas is certainly intriguing but no explanatory discussion or experimental follow-up is offered here. Neither is there a discussion on cerebellar involvement or frontal/ACC. In depth discussion of FUMA outcomes is missing. Possibly rightfully so. But sentences like “physical activity traits share biology with multiple including intelligence, education, obesity, and cardiometabolic diseases” or “to cardiometabolic phenotypes, neurologic health” “showed genetic overlap with neurodegenerative diseases, mental health wellbeing, brain structure, function and connectivity” are uninformative and vague. They also reflect cherry picking because the majority of tested/expected genetic correlations (with CAD or cancer for example) were non-significant.

Authors' response: We gave this particular issue much consideration during the writing of the manuscript. In the above statements we selected the most commonly occurring associations. To further ensure that the interested reader does not interpret these results as ‘cherry picked’, we have made all results available in the supplementary tables and figures. However, the results are currently reported in the current fashion as we decided that readability of the manuscript would be severely affected if we discussed each of the traits that displayed a significant association (n=149 via LD-HUB, n=67 via pheWAS, n>20 via FUMA).

Changes to the paper: None in relation to this comment for the reasons just stated.

Although you invested quite a bit in the MR (directional) analyses on overweight and blood pressure – the discussion does not present clear summary statements of the findings and what they mean. It is all left a bit in the air.

Authors' response: Our MR results advocate the value of physical activity for the reduction of blood pressure. This will of course require replication in other populations with both

genetic and accelerometer-measures of sleep duration and physical activity, as we have stated in the discussion section.

Changes to the paper: Discussion text has been further refined to more clearly reflect that our MR results advocate the value of physical activity for the reduction of blood pressure.

In the abstract you rightfully express concerns that “the etiology of PA and sleep is poorly understood”. How has your work changed this situation? It would be good to make that more visible and explicit. At present I cannot place the remark in the introduction that “the associated loci reveal pathways in the central nervous system that help advance our understanding of biological processes underlying human sleep and activity behaviours”.

Authors’ response: We agree that this sentence could be expressed more accurately. Through the use of heritability partitioning, our study found that tissue and cell enrichment for sleep duration and activity traits point towards a role for the central nervous system. To our knowledge this has not been reported before, and taken together with pathway analyses, thus advances biology knowledge.

Changes to the paper: Taking onboard the reviewer’s concern, we have now changed the relevant introduction section sentence to “*The associated loci reveal pathways in the central nervous system that are associated with sleep duration and activity behaviours*”.

Reviewer 3 MINOR:

-Affiliation 11 is erroneously written as 10.

Authors’ response: Thank you for highlighting this.

Changes to the paper: Typo is now fixed.

-In the abstract, giving the male and female h² estimates would be more informative than the difference ; p-values can be maintained.

Authors’ response: We agree and have now updated the text.

Changes to the paper: Abstract now updated to state “*Heritability was higher in women than in men for overall activity (23% vs. 20%, $p=1.5 \times 10^{-4}$) and sedentary behaviours (18% vs. 15%, $p=9.7 \times 10^{-4}$)*”.

-Page 4, figure S1 should probably be figure 1

Authors' response: Figures have now been updated following removal of accelerometer versus self-report Miami plot.

Changes to the paper: Relevant plot is now removed.

-How many men and women participated? What does the sex difference in SNP-h2 mean? Larger effects for the same genes in females? If so by which mechanism? For $R_g = 0.89$ in MVPA - was this significantly different from 1?

Authors' response: 39,968 men and 51,137 women participated (listed in Figure S4). As detailed in the methods section, heritability differences between men and women for each trait were assessed by extracting a two-tailed p-value from the following z-score (where 'var' indicates variance):

$$z = \frac{h_{females}^2 - h_{males}^2}{\sqrt{\text{var } h_{females}^2 - \text{var } h_{males}^2}}$$

P-values for the estimates are presented in the results section where relevant.

Changes to the paper: None in relation to this comment, as the relevant information is already in the manuscript.

-Table S6-S7 are very difficult to read as they are not organized by accelerometer trait (like table S3). I suggest you switch columns 1 and 3 and sort by accelerometer trait.

Authors' response: We agree and have updated the manuscript accordingly.

Changes to the paper: Relevant tables are updated as suggested.

- Obviously we cannot conclude from tables S6-S9 that all other traits tested (832 - 158 + 4000 - 115 + N_NHGRI - 5) but not found significant, therefore do not share a genetic basis with physical activity or sleep duration. I found this paragraph to generate a lot of concern - the positive correlations can reflect many different causal, mediational or pleiotropic scenario's and the negative ones would need to be qualified by power considerations. But if it must be, I would have preferred the use of MVPA over TPA.

Authors' response: Given many traits were tested, we made a decision to ease readability by just concentrating on the large number that remained after adjustment for multiple testing. As

discussed earlier, we used the overall activity trait rather than the alternative less robust moderate activity trait.

Changes to the paper: None in relation to this comment for the reasons just stated.

- Figure S6 on SNP rs56194509 is unreadable, even viewed at high resolution at my screen. Also, unclear what triangles and color coding mean to convey.

Authors' response: We agree this figure was not in a sufficient resolution. We have therefore decided to instead point the interested reader to supplement table S8.

Changes to the paper: Figure removed.

- Important details on the MR analyses are missing. What is the complete list of traits selected for the MR pipeline in Note S2 – is that what is seen in table S10? Which SNPs were retained as the genetic instrument for the various accelerometer traits?

-Why is there a difference between results in S9 and S10 for e.g. body fat and overall activity? The order of S10 – S9 – S11 – S12ab would have made more sense, I think.

Authors' response: We agree.

Changes to the paper: Order of MR supplement tables is now as suggested.

- Table S10 generates the same question on negative outcomes – surely the absence of MR is not proof of absence of causality, or is it? (Again, I would have preferred the use of MVPA over TPA).

Authors' response: Given many traits were tested, we made a decision to ease readability by just concentrating on those that remained after adjustment for multiple testing. As discussed earlier, we used the overall activity trait rather than the alternative less robust moderate activity trait.

Changes to the paper: None in relation to this comment for the reasons just stated.

- Figure S7 : text printed through the bars is hard to read.

Authors' response: We have updated the labels for S7. We will of course be very happy to work with the production team to further amend figure fonts if necessary.

Changes to the paper: Labels updated in supplement figure S7.

- Methods: how many SNPs were used in genetic association for self-reported PA with the 351,154 participants?

Authors' response: As mentioned at the very start, we now realise a mistake was made in the preparation of the self-reported overall activity phenotype. Given our original paper was already information dense, we now just report results for the accelerometer phenotypes.

Changes to the paper: Removal of analyses comparing self-reported and objective measures of physical activity.

Reviewer #1 (Remarks to the Author):

The authors have improved the presentation of their phenotypes, better distinguishing sleep duration from physical activity estimates, and identified an error in their prior analyses that invalidated the previously presented subjective behavioral reports (now eliminated), thus improving the focus of the manuscript. The authors also have more than doubled the size of their validation cohort and have clarified much of the prior ambiguous or unclear labeling. They also provided explanations for choices of statistical methods/interpretations and presentation decisions. All in all, these findings markedly improve the manuscript.

Despite these improvements, there are areas that could be more specifically addressed, providing a more responsive revision that would assist the reader in understanding the data and the study implications. There are also several areas that would benefit clarification/correction:

Concern about residual population stratification. The authors argue that linear mixed effects models adequately address this concern without a need for additional PC correction. However, the lambdas of 1.20 in settings of p values that are not terribly low do raise concerns that the p values could be somewhat inflated (despite the intercept.) It would be reassuring to see the GWAS findings after additional correction for population stratification. Interestingly, there appeared to be less inflation for the sex stratified analyses vs the full sample for some analyses.

The authors also argue that additional covariate adjustment (for sex, BMI, health status, etc) is not necessary both due to sensitivity analyses and concern about collider adjustments. While I commend concern regarding colliders, that concern certainly does not apply to sex adjustment, which they discuss is a significant covariate and even provide some data supporting variation by sex. BMI adjustment also may assuage concerns not only in regards to potential confounding but as a design variable as accelerometry-based PA may be biased by obesity/BMI (there is mixed data on this, but seeing the adjusted results would be reassuring.)

The authors note that they only have UKB sleep data to compare results. However, CHARGE (Gottlieb) and CARE (Cade, 2016) published sleep duration GWAS and these results should be also referred to.

There remains concerns over the approach to "ground truth." If I understand the methods correctly, "ground truth" for sleep duration was based only on sleep questionnaires/time surveys, and not imaging data. If that is the case, this should be clearly discussed and not referred to as ground truth. The advantages of using acceleratory data trained to a potentially biased data set should be clarified.

If I misunderstood what was done, please clarify the methods for estimating sleep/wake using the n=153 from both imaging and survey data.

The authors still provide little information on their validation set. However, the range of activities and behaviors in that sample as it compares to the larger sample is of interest. It is also important to know the distribution of activities across the day both in the subset and the full data set as some activities will be more or less reliably estimated depending on their representation. (I think that is partially shown in Fig 1 for the full data set--but that Fig is not well described.)

The impact of the results suggesting a causal association between activity and BP is overstated. While PA interventions cannot be blinded, there is wide acceptance due to the consistency of RCTS of a beneficial effect of PA on BP--that is why there are guidelines promoting healthy lifestyle.

Was the p value for a sex interaction 10⁻⁹ overly conservative, especially given heritability differences?

The most novel aspect of this study is the application of ML to the accelerometer data. However, the specifics of this analysis and how the summary data differ from traditional indices were not addressed in any detail. To further the impact of the findings, can the authors compare the distributions/classification using their approach to published data? Are differences in GWAS findings due to unique "signatures" from ML, improved classification, or other differences in statistical methods? In other words, how different is their classification from standard approaches?

The association between sleep is with diastolic BP. However, in the results, this is referred to as systolic BP. (page 3)

The supplemental tables (csv files) did not contain legends for row headings and could be better explained. Table S1 was particularly confusing.

Some inconsistencies in writing remain: e.g. "movement" is sometimes used when referring to both PA and sleep.

Reviewer #2 (Remarks to the Author):

Summary:

In addressing the issues raised by myself and the two other reviewers, it is my opinion that the authors have much improved this manuscript. Whilst the decision to drop the analysis of the self-report measures of activity was no doubt a difficult one, I do believe this has improved the clarity of the manuscript (and that part of the work was admittedly less novel in any case). I have now only a few minor comments to make.

1. P.2 - Introduction – the last sentence of the first paragraph doesn't make sense to me (“..recommend how youth spend”)
2. P.4 – results - there is a reference to Fig. 1 at the end of the first results paragraph – I think this might be a mistake?
3. Fig. S1 (p.4) - The legend of Fig. S1 still refers to self-reported physical activity
4. P.8 – results – you say lowering the p-value to select SNPs for the MR had negligible effects on pleiotropy, did you also check if SNPs (either individually or as a score – where individual level data are available) were associated with any classic confounders, e.g. socioeconomic position, smoking, educational status? Various literature is emerging that suggests latent structure in UK Biobank (whether caused by uncorrected ancestral effects, selection bias/sampling frame, or something else ...) may cause individual SNPs and genetic risk scores (in particular more generous ones) to be associated with such confounders. Establishing that there is no such association is an important step towards validating the assumptions of MR.
5. P.8 – results - In the MR results paragraph and also the MR results tables, units should be added to all the betas (e.g. body mass index (beta per SD higher overall activity: -0.20, SE=0.03, $p=1.1 \times 10^{-10}$) – units of BMI?)
6. Fig S6 (p.8) – there are no single SNP results for hypertension in this figure
7. P.9 – results - The final paragraph of the results refers to Fig. 3 – I cannot find Fig. 3.
8. P.14 – methods - Please define ‘MET’

9. Discussion re. replication – I still think a power calculation here that gives an indication of the predicted size of the replication cohort needed (given the range of effect sizes estimated herein) would be useful.

Reviewer #3 (Remarks to the Author):

The authors have carefully considered most of my comments and used them to change the text of the paper in a number of places. Moreover they conducted additional analyses on the accelerometer data, resolved an error in their previous analyses of the survey measures, and restructured some of the presentations of the results, which –at least for this reviewer- greatly improved the value of the paper.

My earlier positive impression of the paper has therefore been reinforced by this revision. Even so, I remain of a different opinion on a few points:

I noted that the post-GWAS analyses yields an important finding namely evolutionary conserved regions are enriched for PA SNPs. I took this to signal that animal models of the genetics of physical activity (cage movement, voluntary wheel running) might be quite informative for humans, because the variants that influence human PA might also be present in these animals. That is not quite the idea I recognize in the added sentences on page 10. I don't quite see how a human variant that influences PA and sleep in humans "supports a role for PA and sleep throughout mammalian evolution". This would need some additional explanation.

I disagree with the reasoning on systematically favoring TPA over MVPA in some of the finer grained analyses. There is not a shred of evidence to suggest that total physical activity has a larger health impact than MVPA. All guidelines therefore focus on MVPA. It is of course possible that this is the state of play because we had to rely on survey methods, but that is speculation. A counter speculation would be that the 'no pain no gain' adage fully applies and that physiological adaptation occurs only above a moderate to vigorous PA threshold.

The argument that a 100 g threshold is arbitrary is not incorrect per se, but so are many other choices we make in our methodology (say nominal p-value of 0.05 rather than 0.567). As long as one reports what thresholds were used where and when, a study can be replicated independently.

Comments to address (response in red)

We would like to thank the reviewers for their positive and constructive comments on our revised manuscript. We have carefully considered each of the reviewers' comments as reflected in our responses below.

Response to reviewer #1 comments:

Concern about residual population stratification. The authors argue that linear mixed effects models adequately address this concern without a need for additional PC correction. However, the lambdas of 1.20 in settings of p values that are not terribly low do raise concerns that the p values could be somewhat inflated (despite the intercept.) It would be reassuring to see the GWAS findings after additional correction for population stratification. Interestingly, there appeared to be less inflation for the sex stratified analyses vs the full sample for some analyses.

Authors' response: As recognised by the reviewers, we made a number of improvements to the manuscript. One of these was to update the Methods text to state that "Sensitivity analysis on the overall activity trait showed we did not need to include principal components of ancestry as covariates in our linear mixed models ($\lambda=1.20$ v 1.20 , LD intercept = 1.0085 vs 1.0092 , $R_g = 0.99$)." Furthermore, we note that high lambdas in traits with high polygenicity are expected in genome-wide association studies (for example, a recent GWAS in schizophrenia indicated a lambda well above 1.5). We trust that this comparison of lambda and LD intercept comparison information will reassure interested readers who would like to assess the implications of additional correction for population stratification.

Changes to the paper: No changes necessary for reasons just listed.

The authors also argue that additional covariate adjustment (for sex, BMI, health status, etc) is not necessary both due to sensitivity analyses and concern about collider adjustments. While I commend concern regarding colliders, that concern certainly does not apply to sex adjustment, which they discuss is a significant covariate and even provide some data supporting variation by sex. BMI adjustment also may assuage concerns not only in regards to potential confounding but as a design variable as accelerometry-based PA may be biased by obesity/BMI (there is mixed data on this, but seeing the adjusted results would be reassuring.)

Authors' response: We gave this issue much thought before submitting our original manuscript. Following the first set of reviewer comments we subsequently included additional text to the methods section: "*We decided against including too many covariates to avoid the introduction of any possible collider bias that could have an effect on downstream analyses. In addition, we found little difference with the inclusion of: 1) sex as an additional covariate ($R_G=0.99$); 2) BMI as an additional covariate ($R_G=0.93$); and 3) both sex and BMI as additional covariates ($R_G=0.93$).*"

The introduction of sex as a covariate had no effect as indicated in the text above. Furthermore, our sex-stratified analysis "*using LD score regression*²⁶ showed strong

correlation between men and women; correlations ranged from 0.92 for overall activity to 0.97 for sleep duration (Fig S4). In addition, there was no evidence of sexual dimorphism using strict criteria for heterogeneity of effects between men and women ($p < 5 \times 10^{-9}$).”

As noted by the reviewer, it is important to consider the potential for collider bias when adjusting for BMI, particularly as bidirectional effects could be introduced in downstream Mendelian Randomization analyses. In addition, we have been careful in the text not to claim that any potential pathways aren't mediated through BMI or sex. For example, in the discussion section we state that *“It is therefore possible that our findings might be driven through obesity, as we observed strong genetic correlations between activity and obesity traits.”*

In summary, we strongly believe that the current set of confounders {assessment centre, genotyping array, age, age squared, season} are most appropriate for the analysis of accelerometer-measured traits in this manuscript.

Changes to the paper: No changes necessary for reasons just listed.

The authors note that they only have UKB sleep data to compare results. However, CHARGE (Gottlieb) and CARE (Cade, 2016) published sleep duration GWAS and these results should be also referred to.

Authors' response: The CHARGE results were already identified via the GWAS catalog search. This is reflected in Table 1 (see locus 10 near PAX8 and locus 13 near OLFM2) and Table S4 (rs4587207, $p=0.14$ in our data). However, we had not identified the CARE loci, and have now updated the text and table S4 (rs17601612, $p=0.017$) to reflect this.

Changes to the paper: Table S4 updated to show results of locus identified from CARE study and methods text updated to: *“To identify novel loci, we considered index SNPs falling outside 400kb of a SNP previously associated with one of the following: self-report sleep duration or physical activity from the NHGRI-EBI GWAS catalog³⁵, CHARGE²¹ and CARE⁶² consortia; ...”*

There remains concerns over the approach to "ground truth." If I understand the methods correctly, "ground truth" for sleep duration was based only on sleep questionnaires/time surveys, and not imaging data. If that is the case, this should be clearly discussed and not referred to as ground truth. The advantages of using acceleratory data trained to a potentially biased data set should be clarified. If I misunderstood what was done, please clarify the methods for estimating sleep/wake using the $n=153$ from both imaging and survey data.

Authors' response: While the groundtruth is discussed in detail in the methods section, we agree the main concepts could be expressed more clearly in the main body of text.

Changes to the paper: Results text now updated to reflect *“We trained and evaluated the model in 153 free-living individuals (mean age=42, female $n=100$) to distinguish between*

activity states, evaluated against reference wearable camera, time-use, and sleep diary information sources¹⁴”.

Sentence added to discussion section: “To develop machine-learned models of sleep duration in free living environments, we did not use traditional cumbersome polysomnography methods, and instead relied on self-reported sleep diaries which can be prone to measurement error.”

The authors still provide little information on their validation set. However, the range of activities and behaviors in that sample as it compares to the larger sample is of interest. It is also important to know the distribution of activities across the day both in the subset and the full data set as some activities will be more or less reliably estimated depending on their representation. (I think that is partially shown in Fig 1 for the full data set--but that Fig is not well described.)

Authors’ response: While the validation dataset is described in much detail in a recent Scientific Reports paper (*Willets et al*), we agree it would be helpful to show the distribution of activities across the day both in the training dataset (n=153) and the UK Biobank dataset (n=91,105). As expected there are small differences in the distribution of activities, but as noted in the discussion section “...the size of our training dataset has helped provide a diverse set of representative behaviours, rather than individuals, which is important for model development¹⁴”.

Changes to the paper: New supplement figure S11 created: “Distribution of groundtruth activity states in training dataset (n=153) and predicted labels in UK Biobank accelerometer dataset (n=91,105).” Text added to methods section stating: “An overview of the distribution of activity behaviours in this training dataset is provided in Figure S11.”

The impact of the results suggesting a causal association between activity and BP is overstated. While PA interventions cannot be blinded, there is wide acceptance due to the consistency of RCTS of a beneficial effect of PA on BP--that is why there are guidelines promoting healthy lifestyle.

Authors’ response: We respectfully disagree with this comment, as we have used cautious wording in relation to the MR blood pressure findings e.g. abstract text “...suggest that increased overall activity might be causally related to lowering diastolic blood pressure...”. Despite a consistency of small RCTs showing a short-term beneficial effect of physical activity on blood pressure, we stand by our statement that past findings could be prone to ascertainment bias due to difficulties in blinding participants to behavioural interventions.

Changes to the paper: No changes necessary for reasons just listed.

Was the p value for a sex interaction 10^{-9} overly conservative, especially given heritability differences?

Authors' response: This was a genome-wide test of almost 10 million SNPs, across five traits (overall activity, sleep duration, sedentary, walking, moderate) therefore a threshold of $p < 5 \times 10^{-9}$ is necessary to mitigate against false positive chance findings.

Changes to the paper: No changes necessary for reasons just listed.

The most novel aspect of this study is the application of ML to the accelerometer data. However, the specifics of this analysis and how the summary data differ from traditional indices were not addressed in any detail. To further the impact of the findings, can the authors compare the distributions/classification using their approach to published data? Are differences in GWAS findings due to unique "signatures" from ML, improved classification, or other differences in statistical methods? In other words, how different is their classification from standard approaches?

Authors' response: We agree it would help the reader to know how the machine learning phenotypes differ from traditional overall activity. To the best of our knowledge, our recent Scientific Reports paper (*Willettts et al*) is the only application of machine-learned models of activity behaviour in a population scale dataset. Therefore, the only comparison available is against traditional overall activity metrics, which we now show via phenotype and genotype correlation matrices.

Changes to the paper: Phenotype and genotype correlation matrices are added in new supplement figure S10, and results text has been updated to state that "*The model was applied to over 100,000 UK Biobank participants¹⁴ (Fig 1), where the machine learned phenotypes provided orthogonal information to the traditional overall activity time phenotype (Fig S10).*"

The association between sleep is with diastolic BP. However, in the results, this is referred to as systolic BP. (page 3)

Authors' response: We are unable to find where this was stated in the manuscript. However, we have now removed previously reported associations between sleep duration and hypertension, following confounding (with education years), multivariate, and univariate (IVW) Mendelian Randomization analyses (table S12B). For further information, please refer to our detailed response to a comment raised by reviewer 2 on MR sensitivity analyses.

Changes to the paper: Sleep duration associations with hypertension removed from abstract, results, and Tables S13A & S13B.

The supplemental tables (csv files) did not contain legends for row headings and could be better explained. Table S1 was particularly confusing.

Authors' response: We have carefully reviewed headings for all supplement tables and agree that adding a legend to Table S1 would be beneficial.

Changes to the paper: Legend now added to explain column headings in Table S1: *“Phenotype: Phenotype of interest ... Wearable camera annotation: Annotation by researcher viewing only wearable camera images ... MET: Metabolic equivalent of task - an indication of how intense a task is (<=1 indicates being at rest) ... Compendium of physical activities code: A code to link camera annotation information to activity intensity information”*

Some inconsistencies in writing remain: e.g. "movement" is sometimes used when referring to both PA and sleep.

Authors' response: Thank you for this feedback.

Changes to the paper: All instances of “movement” phenotypes are now updated to “accelerometer-defined” phenotypes.

Response to reviewer #2 comments:

1. P.2 - Introduction – the last sentence of the first paragraph doesn't make sense to me (“..recommend how youth spend”)

Authors' response: We agree this could be more clearly worded.

Changes to the paper: Sentence now updated to *“In response, recent Canadian public health guidelines recommend how much physical activity and sleep duration youths should engage in for health benefit⁶”*

2. P.4 – results - there is a reference to Fig. 1 at the end of the first results paragraph – I think this might be a mistake?

Authors' response: We have double-checked this reference and believe it should remain. This figure has two purposes, 1) to show the distribution of phenotypes identified by the machine-learned model; and 2) to list SNPs associated with these machine-learned phenotypes.

Changes to the paper: No changes necessary for reasons just listed.

3. Fig. S1 (p.4) - The legend of Fig. S1 still refers to self-reported physical activity

Authors' response: We thank the reviewer to identifying this labelling error.

Changes to the paper: Fig S1 legend now updated to *“Summary Manhattan and QQ plots of European sex-combined GWAS of accelerometer-measured physical activity and sleep traits: the UK Biobank study 2013-2015 (n = 91,105). Genomic control (λ), explained variance (R^2) and heritability (h^2) estimates are also provided.”*

4. P.8 – results – you say lowering the p-value to select SNPs for the MR had negligible effects on pleiotropy, did you also check if SNPs (either individually or as a score – where individual level data are available) were associated with any classic confounders, e.g. socioeconomic position, smoking, educational status? Various literature is emerging that suggests latent structure in UK Biobank (whether caused by uncorrected ancestral effects, selection bias/sampling frame, or something else ...) may cause individual SNPs and genetic risk scores (in particular more generous ones) to be associated with such confounders. Establishing that there is no such association is an important step towards validating the assumptions of MR.

Authors' response: We agree that it would be good to check if SNPs are associated with classic confounders. We have therefore added a new supplement table (S12B) to conduct these sensitivity analyses on the following potential confounders: income, smoking, area deprivation, and years of education. A potential association with years of education was found, but a follow-up multivariate mendelian randomization analysis indicated that causal

inference can be permitted for overall activity even if no variants are uniquely associated with just activity alone. However, our previously reported association between sleep duration and hypertension is now removed, following multivariate and univariate (IVW) MR analyses (table S12B).

Changes to the paper: New supplement table created, S12B “*Confounding sensitivity analyses for Mendelian Randomization associations...*” Sleep duration associations with hypertension removed from abstract, results, and Tables S13A & S13B. Methods text updated to include “*Sensitivity analyses was also conducted to test whether instrument variables were associated with the following observational ‘classic’ confounders: income, smoking, area deprivation, and years of education*⁷⁵. *Potential associations with confounders were subsequently followed-up by multivariate mendelian randomization analyses*⁷⁶ *to investigate whether causal inference could be permitted even if variants were not uniquely associated with just the trait of interest.*”

5. P.8 – results - In the MR results paragraph and also the MR results tables, units should be added to all the betas (e.g. body mass index (beta per SD higher overall activity: -0.20, SE=0.03, p=1.1x10⁻¹⁰) – units of BMI?)

Authors’ response: We have updated the abstract and results text as suggested, and also added in columns to denote units for all MR tables.

Changes to the paper: Tables S10-13 updated to include column to denote units. In addition, abstract sentence updated to: “*...increased overall activity might be causally related to lowering diastolic blood pressure (beta mmHg per SD: -0.95, SE=0.19, p=9.5x10⁻⁷)...*”

Results sentence updated to: “*We found consistent evidence of inverse relationships for overall activity with body mass index (beta units of BMI per SD higher overall activity: -0.20, SE=0.03, p=1.1x10⁻¹⁰), diastolic blood pressure (beta mmHg per SD: -0.95, SE=0.19, p=9.5x10⁻⁷)...*”

6. Fig S6 (p.8) – there are no single SNP results for hypertension in this figure

Authors’ response: As recognised by the reviewers, we made a number of improvements to the manuscript. One of these is the inclusion of Mendelian randomisation leave-one-out and single-SNP-only sensitivity analyses for traits suggesting evidence of causality, colour-coded by p-value for instrument variable in discovery GWAS (Fig S6). This figure includes single SNP results for hypertension (bottom two images on first page of figure).

Changes to the paper: No changes necessary for reasons just listed.

7. P.9 – results - The final paragraph of the results refers to Fig. 3 – I cannot find Fig. 3.

Authors’ response: We thank the reviewer for pointing out this typo.

Changes to the paper: Reference to Fig 3 deleted, and instead updated to correctly reference Fig S7.

8. P.14 – methods - Please define ‘MET’

Authors’ response: Thank you for noting that this was not explicitly defined on first mention.

Changes to the paper: Methods section sentence now updated to: “*A sedentary behaviour is one which has a MET (Metabolic Equivalent of Task) energy expenditure score...*”

9. Discussion re. replication – I still think a power calculation here that gives an indication of the predicted size of the replication cohort needed (given the range of effect sizes estimated herein) would be useful.

Authors’ response: We have now added a power calculation as suggested.

Changes to the paper: Sentence added to discussion section: “*The high polygenicity of these traits makes it difficult to precisely estimate replication sample sizes. Assuming the replication p-value is set at $0.05/15$ loci = 3.3×10^{-3} , and variance explained by a SNP ranges between $1.5 - 5.0 \times 10^{-4}$, we estimate between ~36k and ~119k participants are necessary to provide 90% power to replicate the loci listed in **Table 1.***”

Response to reviewer #3 comments:

I noted that the post-GWAS analyses yields an important finding namely evolutionary conserved regions are enriched for PA SNPs. I took this to signal that animal models of the genetics of physical activity (cage movement, voluntary wheel running) might be quite informative for humans, because the variants that influence human PA might also be present in these animals. That is not quite the idea I recognize in the added sentences on page 10. I don't quite see how a human variant that influences PA and sleep in humans "supports a role for PA and sleep throughout mammalian evolution". This would need some additional explanation.

Authors' response: We mostly agree with this comment and have just one slight disagreement. Firstly, we agree that our heritability partitioning analysis indicates that animal models of physical activity genetics might be quite informative for humans. However, one could argue that a variant that influences PA and sleep in humans, which is in a region that happens to be conserved across mammals, indicates some role for activity and sleep duration in mammalian evolution.

Changes to the paper: Discussion sentence updated to: "*Animal models might be informative to answer such questions, as our heritability partitioning analysis supports a role for physical activity and sleep behaviors in regions conserved in mammals*".

I disagree with the reasoning on systematically favoring TPA over MVPA in some of the finer grained analyses. There is not a shred of evidence to suggest that total physical activity has a larger health impact than MVPA. All guidelines therefore focus on MVPA. It is of course possible that this is the state of play because we had to rely on survey methods, but that is speculation. A counter speculation would be that the 'no pain no gain' adage fully applies and that physiological adaptation occurs only above a moderate to vigorous PA threshold.

The argument that a 100 g threshold is arbitrary is not incorrect per se, but so are many other choices we make in our methodology (say nominal p-value of 0.05 rather than 0.567). As long as one reports what thresholds were used where and when, a study can be replicated independently.

Authors' response: As recognised by the reviewers, we made a number of improvements to the manuscript. One was a new definition of moderate activity that no longer overlapped with other traits, and offered concordant information to the overall activity variable. Therefore, we have included fine grained analysis on all traits (including moderate activity) where possible, such as heritability partitioning, sexual dimorphism, genetic correlation, PheWAS, Mendelian Randomisation, and pathway analysis. In some instances it was not possible to conduct analysis on walking and moderate traits as they did not have loci that reached genome wide significance ($p < 5 \times 10^{-9}$), such as for finemapping and gene set enrichment analysis.

Changes to the paper: Figure S8 updated to state that no significant loci were available on which to conduct gene set enrichment analysis.

Reviewer #1 (Remarks to the Author):

Thank you for considering the comments provided before.

Despite the arguments provided, I believe the readership would benefit from seeing the actual associations and p values for sex and BMI-adjusted models for the reasons stated previously-- actigraphy data can provide varied information on activity/sleep by sex and BMI, and a minimal sensitivity analysis would be helpful to readers interested in the genetics of accelerometry-based phenotypes.

Figure S10 is shown to describe the "orthogonal" information provided by ML vs traditional labeled accelerometry-- but without better labeling, it looks like different dimensions of sleep/activity are being correlated-but not clearly showing associations among different sources of that information. My comment was that given that the ML output is used to identify novel loci, it would be useful to more explicitly describe how the activity/sleep output from ML differed from that in the UKBiobank and discuss whether the results of this study were driven by different activity/sleep phenotypes vs differences in sample/modelling techniques. As noted in the prior review, Figure S11B seems to get at the differences in phenotype by the two methods, but this figure does not quantify how different the overall distributions of these measures are compared to UKBiobank.

On page 3, you write, " Our study also reveals the importance of having objective measures at scale to detect genetic effects for behavioural traits. Mendelian randomization supports a role for physical activity in the reduction of body fat percentage and systolic blood pressure." However, you highlighted findings with diastolic blood pressure.

Reviewer #2 (Remarks to the Author):

Thank you for your thorough response to my last review. I have just a few final queries, as follows.

P.2, "Emerging evidence shows ..." – odd wording here

P.6, "Using heritability estimates ..." – this sentence references Fig. S2 – should this be S1?

P.15, “Sensitivity analysis ...” – please could you define R_g/R_G – is this the correlation of betas/P-values or the genetic correlation (in which case how has this been calculated?). Also, how many PCs were fitted? And when you talk about the LD intercept – presumably this is from LD score regression – should you be referencing Bulik-Sullivan?

Fig S10 – could the figure title be modified to make it clearer that the principal comparison here is intended to be between the traditionally derived ‘overall activity’ metric and the four machine-learning derived measures?

Suppl. Table S12B – please could you clarify what is shown in the column called ‘Accelerometer trait (univariate IVW comparison)’ – this doesn’t seem to be the result from any of the other MR tables.

Regarding added text “beta units of BMI per SD higher overall activity” – in this case I think it’s in terms of SD of BMI isn’t it? Not sure what ‘beta units’ means.

Fig. S6 – Sorry, I am still confused by this - why is there one ‘extra’ Hypertension - UKB figure in the ‘Leave one SNP out’ column (bottom of page 2 of this figure). Is this for a different phenotype (sleep duration) as it shows an $OR > 1$?

Response to reviewer #1 comments:

Despite the arguments provided, I believe the readership would benefit from seeing the actual associations and p values for sex and BMI-adjusted models for the reasons stated previously-- actigraphy data can provide varied information on activity/sleep by sex and BMI, and a minimal sensitivity analysis would be helpful to readers interested in the genetics of accelerometry-based phenotypes.

Authors' response: We have now updated table S3 to include sensitivity analyses with sex and BMI as additional covariates. The strength of association appears attenuated for two overall activity loci (rs564819152, rs59499656), one sleep duration locus (rs13002630), one sedentary behaviour locus (rs25981), and one moderate intensity activity locus (rs10849145). Taking potential collider bias into account, understanding the implication of these attenuations is clearly complex and we have been careful in the text not to claim that any potential pathways aren't mediated through BMI or sex. For example, in the discussion section we state: *"It is therefore possible that our findings might be driven through obesity, as we observed strong genetic correlations between activity and obesity traits. However, as this is one of the first large-scale physical activity genetic studies, we have likely only begun to understand the complex interplay between activity, cardiometabolic phenotypes, and neurologic health."*

Given the potential for collider bias when adjusting for BMI, bidirectional effects could be introduced in downstream Mendelian Randomization analyses. Therefore, we strongly believe that the current set of confounders {assessment centre, genotyping array, age, age squared, season} are most appropriate for the analysis of accelerometer-measured traits in this manuscript.

Changes to the paper: Table S3 now updated to include sensitivity analyses with sex and BMI as additional covariates. Methods text updated to include sentence: *"Table S3 reports a sensitivity analysis for each locus with sex and BMI included as additional covariates, the results of which should be cautiously interpreted due to the potential for collider bias when adjusting for BMI."*

Figure S10 is shown to describe the "orthogonal" information provided by ML vs traditional labeled accelerometry-- but without better labeling, it looks like different dimensions of sleep/activity are being correlated-but not clearly showing associations among different sources of that information. My comment was that given that the ML output is used to identify novel loci, it would be useful to more explicitly describe how the activity/sleep output from ML differed from that in the UKBiobank and discuss whether the results of this study were driven by different activity/sleep phenotypes vs differences in sample/modelling techniques. As noted in the prior review, Figure S11B seems to get at the differences in phenotype by the two methods, but this figure does not quantify how different the overall distributions of these measures are compared to UKBiobank.

Authors' response: We interpret there to be two main issues raised in this long comment:

1) What is the purpose of Figure S10?

Based on the helpful suggestion of reviewer 2, we have updated the caption of Figure S10 to more clearly reflect that the principal comparison here is between the traditionally derived ‘overall activity’ metric and the four machine-learning derived measures. This shows that the ML derived phenotypes offer orthogonal information to traditionally derived accelerometry with correlations ranging from -0.31 (sleep) to 0.48 (moderate activity).

Please note that the term ‘traditional **labeled** accelerometry’ is inaccurate as the overall activity metric is a scalar value where lower values indicate low movement and higher values indicate more movement/activity but offer no insight on the *behavioural type* of activity. The only way to currently interpret objectively-measured physical activity *behaviour type* information in UK Biobank is via the method we report in this paper.

2) Are the ML phenotypes derived from the n=153 training dataset sufficiently robust for population inference in n=91,105 UK Biobank participants?

Please refer to response on page one to editor’s comment on this issue, where we show that the ML-derived phenotypes have been trained on a sufficiently diverse set of representative behaviours to ensure their validity for use in UK Biobank.

Changes to the paper: Figure S10 caption updated to: “**Figure S10** | Phenotypic and genetic correlation between traditional ‘overall activity’ metric and four machine-learned phenotypes in 91,105 UK Biobank participants who wore wrist-worn accelerometers.”.

New figure added: **Figure S9** | Phenotype face validity check: difference in accelerometer-measured sleep classification by self-reported chronotype in 91,105 UK Biobank participants.

Results text now updated to state: “...This labelled free-living dataset is many times larger than equivalent laboratory datasets (188,355 vs. ~3,600 minutes of annotated behaviour)¹⁴ and thus provide a diverse set of representative free-living behaviours. Our model achieved an overall classification score of $\kappa=0.68$ in correctly predicting what activity state an individual is in for any given 30-second time period (**Table S2**). As detailed in a previous publication¹⁴ this model was considered suitable for population inference in UK Biobank as the performance was not materially affected by sex and age (difference in κ score: $\text{sex} < 0.0001$, $\text{age} < 0.05$). Therefore, our model was applied to over 100,000 UK Biobank participants (**Fig 1**), where the machine learned phenotypes demonstrated face validity¹⁴ with clear and expected differences in sleep duration by self-reported chronotype status (**Fig S9**). These phenotypes provided orthogonal information to the traditional overall activity time phenotype (**Fig S10**)...”

On page 3, you write, " Our study also reveals the importance of having objective measures at scale to detect genetic effects for behavioural traits. Mendelian randomization supports a role for physical activity in the reduction of body fat percentage and systolic blood pressure." However, you highlighted findings with diastolic blood pressure.

Authors' response: We thank the reviewer for highlighting this error in the text.

Changes to the paper: Text updated to: "*Mendelian randomization supports a role for physical activity in the reduction of body fat percentage and diastolic blood pressure.*"

Response to reviewer #2 comments:

1. P.2, "Emerging evidence shows ..." – odd wording here

Authors' response: We have updated this wording.

Changes to the paper: Sentence now updated to "*Recent GWAS studies have indicated a genetic contribution to physical activity,*"

P.6, "Using heritability estimates ..." – this sentence references Fig. S2 – should this be S1?

Authors' response: We thank the reviewer for highlighting this error in the text.

Changes to the paper: Text updated to: "*Using heritability estimates from BOLT-LMM¹⁵ (Methods), we found all traits to have modest heritability, ranging from 10% for moderate intensity activity to 21% for overall activity (Fig S1).*"

P.15, "Sensitivity analysis ..." – please could you define R_g/R_G – is this the correlation of betas/P-values or the genetic correlation (in which case how has this been calculated?). Also, how many PCs were fitted? And when you talk about the LD intercept – presumably this is from LD score regression – should you be referencing Bulik-Sullivan?

Authors' response: We have now updated the text to reflect that R_g refers to the genetic correlation calculated via the LD score regression method of Bulik-Sullivan, showing that no PCs were fitted.

Changes to the paper: Text updated to: "*Sensitivity analysis using LD score regression²⁶ on the overall activity trait showed we did not need to include principal components of ancestry as covariates in our linear mixed models ($\lambda=1.20$ v 1.20 , LD intercept = 1.0085 vs 1.0092 , genome-wide genetic correlation (R_g) = 0.99)."*

Fig S10 – could the figure title be modified to make it clearer that the principal comparison here is intended to be between the traditionally derived ‘overall activity’ metric and the four machine-learning derived measures?

Authors’ response: As indicated in our response to reviewer 1, we agree with reviewer 2 that the caption for Figure S10 should be improved.

Changes to the paper: Figure S10 caption updated to: “**Figure S10** | Phenotypic and genetic correlation between traditional ‘overall activity’ metric and four machine-learned phenotypes in 91,105 UK Biobank participants who wore wrist-worn accelerometers.”

Suppl. Table S12B – please could you clarify what is shown in the column called ‘Accelerometer trait (univariate IVW comparison)’ – this doesn’t seem to be the result from any of the other MR tables.

Authors’ response: To assess whether causal inference can be permitted for a trait even if no variants are uniquely associated with it, we used the multivariate mendelian randomization method developed by Burgess and colleagues (reference 76). This method relies on inverse variance weighting (IVW). We believe it is helpful to place the multivariate result in context by providing a univariate IVW comparison.

We would not expect this sensitivity result to match results in the other MR tables as our main analyses relies on the maximum likelihood method rather than IVW.

Changes to the paper: No changes necessary for reasons just listed.

Regarding added text “beta units of BMI per SD higher overall activity” – in this case I think it’s in terms of SD of BMI isn’t it? Not sure what ‘beta units’ means.

Authors’ response: We thank the reviewer for highlighting this error in the text.

Changes to the paper: Text updated to: “We found consistent evidence of inverse relationships for overall activity with body mass index (beta SD of BMI per SD higher overall activity...”

Fig. S6 – Sorry, I am still confused by this - why is there one ‘extra’ Hypertension - UKB figure in the ‘Leave one SNP out’ column (bottom of page 2 of this figure). Is this for a different phenotype (sleep duration) as it shows an OR > 1?

Authors’ response: We are sorry for the confusion here, as we mistakenly included a legacy hypertension figure for the sleep phenotype. This is now removed.

Changes to the paper: Figure S6 now updated to remove the unnecessary hypertension graphic.

Reviewer #1 (Remarks to the Author):

The authors provide additional clarification regarding concerns raised.

In response to the request to the concern of how well the in-lab data set reflected the behaviors in the targeted data set, the authors responded by noting the large size of this data set. However, using "minutes" of data is somewhat deceptive. (This labelled free-living dataset is many times larger than equivalent laboratory datasets (188,355 vs. ~3,600 minutes of annotated behaviour) 14 and thus provides a diverse set of representative free-living behaviours.") These minutes are not independent are one another and the concern is whether the samples engaged in a similar range and characteristic type of activities across the two datasets. I suggest this new defensive sentence be eliminated and the authors more simply acknowledge this potential limitation.

In response to the request to show sex and BMI-adjusted results, these are now shown in a supplementary table. Note that the pdf version does not display the columns of information. In the CSV file, however, what is seen is that not only do several p values increase, but some beta estimates actually reverse directionality. Rather than simply dismiss these changes as due to collider bias (which could not be argued if sex adjustment accounted to such changes), the authors could more thoughtfully discuss the implications of the altered results when considering sex and BMI adjustment. They may need to see what adjustments drive the changes in estimates and statistical significance. If it is BMI and not sex, a brief discussion of this would be helpful, especially since the evidence for collider bias is arguable.

Reviewer #2 (Remarks to the Author):

Thank you for your response. I am satisfied with your changes and have no further comments to add.

Response to reviewer #1 comments:

In response to the request to the concern of how well the in-lab data set reflected the behaviors in the targeted data set, the authors responded by noting the large size of this data set. However, using "minutes" of data is somewhat deceptive. (This labelled free-living dataset is many times larger than equivalent laboratory datasets (188,355 vs. ~3,600 minutes of annotated behaviour) 14 and thus provides a diverse set of representative free-living behaviours.") These minutes are not independent are one another and the concern is whether the samples engaged in a similar range and characteristic type of activities across the two datasets. I suggest this new defensive sentence be eliminated and the authors more simply acknowledge this potential limitation.

Authors' response: We are happy to remove the text on minutes of behaviour, and also retain our discussion section sentence acknowledging that the participants in our machine learning free-living training dataset are not a random subsample of the UK Biobank study.

Changes to the paper: Relevant sentence removed from results subsection "*Statistical machine learning of behaviours from sensor data*". For reference this deleted sentence was "*This labelled free-living dataset is many times larger than equivalent laboratory datasets (188,355 vs ~3,600 minutes of annotated behaviour)¹⁴ and thus provides a diverse set of representative free-living behaviours*"

In response to the request to show sex and BMI-adjusted results, these are now shown in a supplementary table. Note that the pdf version does not display the columns of information. In the CSV file, however, what is seen is that not only do several p values increase, but some beta estimates actually reverse directionality.

Authors' response: We would like to clarify that the beta estimates do not reverse for any locus after adjusting for sex, BMI, or sex+BMI, (see Supplementary Data 2).

Changes to the paper: Supplement table now converted into Supplementary Data 2.

Rather than simply dismiss these changes as due to collider bias (which could not be argued if sex adjustment accounted to such changes), the authors could more thoughtfully discuss the implications of the altered results when considering sex and BMI adjustment. They may need to see what adjustments drive the changes in estimates and statistical significance. If it is BMI and not sex, a brief discussion of this would be helpful, especially since the evidence for collider bias is arguable.

Authors' response: For this final revision, we have now updated Supplementary Data 2 to include sensitivity analyses with the following additional covariates: i) sex, ii) BMI, iii) sex and BMI. These results indicate that all attenuations in strength of association for candidate

loci were driven by BMI alone, and not sex. However, we agree with the reviewer and editor that understanding the implication of these attenuations is clearly complex. Therefore, we have been careful in the text not to claim that any potential pathways aren't mediated through BMI or sex.

Changes to the paper: Supplementary Data 2 now updated to include sensitivity analyses with the following additional covariates: i) sex, ii) BMI, iii) sex and BMI.

Methods text updated to include sentence: *“Supplementary Data 2 reports a sensitivity analysis for each locus with sex and BMI included as additional covariates, where all attenuations were driven by BMI alone. These results should be cautiously interpreted due to the potential for collider bias when adjusting for BMI.”*

Discussion text updated to include sentence: *“It is therefore possible that our findings might be driven through obesity, as we observed strong genetic correlations between activity and obesity traits, and the strength of association for some activity loci was attenuated when including BMI as an additional covariate. However, we have likely only begun to understand the complex interplay between activity, cardiometabolic phenotypes, and neurologic health.”*

Response to reviewer #2 comments:

Thank you for your response. I am satisfied with your changes and have no further comments to add.

Authors' response: Thank you to all the reviewers for improving this manuscript throughout the entire revision process.

Reviewer #2

I am broadly happy with the changes made in response to Reviewer #1's last set of comments. However, I would like to request an explanation as to why some results have been changed in this version (see details below) and have identified a couple of other minor points (listed below).

Updated results:

As well as the changes made in response to Reviewer #1's comments, it seems that there have been some changes to the results in the current version (as summarised below). The changes are in most cases small but there are some that are more significant (e.g. change in no. of GW significant loci from 12 to 13; change in one of the SNPs in Table 1). The authors do not seem to provide any indication as to why these changes have been made and there appears to be no corresponding updates in the methods (indicating, for example, that there has been a change in the no. of individuals in the analysis). Please could the authors explain the reason for these changes? Have these analyses been rerun with new withdrawals of consent (in UKBB) or with updated databases (e.g. FUMA), for example?

- Two-sample Mendelian randomization suggests that increased activity might causally lower diastolic blood pressure (beta mmHg/SD: -0.91, SE=0.18, $p=8.2 \times 10^{-7}$), and odds of hypertension (Odds ratio/SD: 0.84, SE=0.03, $p=4.9 \times 10^{-8}$).
- Setting genome-wide significance at the conventional threshold of $p < 5 \times 10^{-8}$ would have revealed 13 [was 12] additional loci, 11 [was 10] of which are novel (Supplementary Data 2).
- Gene-based analysis using FUMA28, with input SNPs mapped by position to 18,232 protein-coding genes, implicated an additional locus for walking (near HP1BP3) and moderate activity (near KCNA6) (Supplementary Data 4). However, no further loci were found to those already identified in the SNP association analysis.
- Genome-wide significant loci account for 0.06% [was <0.01%] (overall activity) to 0.33% (sleep duration) of phenotypic variance across the traits (Supplementary Figure 3).
- Three loci contained plausible causal variants in regions spanning <15kb (Supplementary Table 2).
- We found consistent evidence of inverse relationships for overall activity with body mass index (beta SD of BMI per SD higher overall activity: -0.14, SE=0.015, $p=8.7 \times 10^{-20}$ [was $p=2.4 \times 10^{-16}$]), diastolic blood pressure (beta mmHg per SD: -0.91, SE=0.18, $p=8.2 \times 10^{-7}$), and hypertension (Odds ratio per SD: 0.84, SE=0.03, $p=4.9 \times 10^{-8}$).

- While these loci alone account for less than 0.39% of phenotypic variation, our analyses suggest that as much as 18% [up from 9%] of physical activity and sleep duration variation might be accounted for by common (minor allele frequency > 1%) genetic variation.
- Here, we found brain tissues to be enriched ($p < 1.8 \times 10^{-4}$), in particular at the cerebellum, frontal cortex, brain cortex, and anterior cingulate cortex for at least two [was three] accelerometer traits (Supplementary Figure 9).
- Table 1 – rs7765476 replaced with rs72828533

Minor issues:

1. Figure 1 – what are the *'s indicating?
2. Table 1 and Supp. Data 2 – in Table 1 rs7765476 has been replaced with rs72828533 but rs7765476 is still in Suppl. Data 2.
3. Suppl. Data 3 – there are two worksheets in this file – is this intended?
4. I am confused by the statement as it now seems contradictory with respect to whether additional loci were found or not: “Gene-based analysis using FUMA28, with input SNPs mapped by position to 18,232 protein-coding genes, implicated an additional locus for walking (near HP1BP3) and moderate activity (near KCNA6) (Supplementary Data 4). However, no further loci were found to those already identified in the SNP association analysis.”
5. Supplementary Table 12 – typo ‘accelerometer’ in first yellow box

Response to Reviewer #2 comments

We identified a minor implementation bug in the preparation of our genotype data. As noted by the reviewer, the changes to our results are in most cases small. While these changes are minor, the new data are based on more accurate representation of the methods described in the manuscript. Therefore, the methods text in our manuscript remains unchanged.

The bug occurred during quality control of the BOLT-LMM output, where we used PLINK (version 1.9) to exclude candidate variants ($p < 1 \times 10^{-5}$) that deviated from Hardy-Weinberg equilibrium ($p < 1 \times 10^{-7}$). Computationally, we managed this as a cluster process where the job was split into many smaller components/regions. We discovered an implementation bug where a very small number of those jobs silently failed due to a memory leak. Therefore, while the data in our summary stats file was all correct, the bug resulted in a small number of chunks that should have been in the summary stats file but were missing. Fixing this bug had an effect on subsequent analyses such as loci identification, genetic correlation analyses, etc. The overall findings for our manuscript remain the same as reported before.

We would like to thank the reviewer for their diligence and inline below (in red font) address how the implementation bug affected each of the changes noted by reviewer #2.

=== End of response to Reviewer #2 comments ===

Reviewer #2:

I am broadly happy with the changes made in response to Reviewer #1's last set of comments. However, I would like to request an explanation as to why some results have been changed in this version (see details below) and have identified a couple of other minor points (listed below).

Updated results:

As well as the changes made in response to Reviewer #1's comments, it seems that there have been some changes to the results in the current version (as summarised below). The changes are in most cases small but there are some that are more significant (e.g. change in no. of GW significant loci from 12 to 13; change in one of the SNPs in Table 1). The authors do not seem to provide any indication as to why these changes have been made and there appears to be no corresponding updates in the methods (indicating, for example, that there has been a change in the no. of individuals in the analysis). Please could the authors explain the reason for these changes? Have these analyses been rerun with new withdrawals of consent (in UKBB) or with updated databases (e.g. FUMA), for example?

Authors: See overall response above.

- Two-sample Mendelian randomization suggests that increased activity might causally lower diastolic blood pressure (beta mmHg/SD: -0.91, SE=0.18, $p=8.2 \times 10^{-7}$), and odds of hypertension (Odds ratio/SD: 0.84, SE=0.03, $p=4.9 \times 10^{-8}$).

Authors: In our updated analysis, some additional instrument variables were identified to support Mendelian Randomization analyses, where for example 68 (rather than 62) instruments should be included.

- Setting genome-wide significance at the conventional threshold of $p < 5 \times 10^{-8}$ would have revealed 13 [was 12] additional loci, 11 [was 10] of which are novel (Supplementary Data 2).

Authors: After fixing the implementation bug, the area around rs6775319 in chromosome 18 could be included in the analysis and identified an additional locus as being potentially associated with overall activity. None of the other 'missing chunks' contained loci that passed that conventional threshold of $p < 5 \times 10^{-8}$.

- Gene-based analysis using FUMA28, with input SNPs mapped by position to 18,232 protein-coding genes, implicated an additional locus for walking (near HP1BP3) and moderate activity (near KCNA6) (Supplementary Data 4). However, no further loci were found to those already identified in the SNP association analysis.

Authors: The inclusion of additional QCed data resulted in small changes to point estimates for gene-based analysis using FUMA, where the input file now contains a complete summary stats file.

- Genome-wide significant loci account for 0.06% [was $< 0.01\%$] (overall activity) to 0.33% (sleep duration) of phenotypic variance across the traits (Supplementary Figure 3).

Authors: The changes in phenotypic variance estimates are entirely explained by an update to the most-up-to-date PRSice software.

- Three loci contained plausible causal variants in regions spanning $< 15\text{kb}$ (Supplementary Table 2).

Authors: As none of the main loci ($p < 5 \times 10^{-9}$) were affected by the implementation bug, fine-mapping results did not change. However, the previous versions of our manuscript wrongly reported three loci containing plausible causal variants in regions spanning $< 10\text{kb}$ (this should be $< 15\text{kb}$, and the text was updated to reflect this).

- We found consistent evidence of inverse relationships for overall activity with body mass index (beta SD of BMI per SD higher overall activity: -0.14 , $SE=0.015$, $p=8.7 \times 10^{-20}$ [was $p=2.4 \times 10^{-16}$]), diastolic blood pressure (beta mmHg per SD: -0.91 , $SE=0.18$, $p=8.2 \times 10^{-7}$), and hypertension (Odds ratio per SD: 0.84 , $SE=0.03$, $p=4.9 \times 10^{-8}$).

Authors: As mentioned earlier, some additional loci were identified using less strict criteria ($p < 5 \times 10^{-6}$) to support Mendelian Randomization analyses, where for example 68 (rather than 62) instruments should be included.

- While these loci alone account for less than 0.39% of phenotypic variation, our analyses suggest that as much as 18% [up from 9%] of physical activity and sleep duration variation might be accounted for by common (minor allele frequency $> 1\%$) genetic variation.

Authors: As mentioned earlier, the changes in phenotypic variance estimates are entirely explained by an update to the most-up-to-date PRSice software.

- Here, we found brain tissues to be enriched ($p < 1.8 \times 10^{-4}$), in particular at the cerebellum, frontal cortex, brain cortex, and anterior cingulate cortex for at least two [was three] accelerometer traits (Supplementary Figure 9).

Authors: The inclusion of additional QCed data resulted in the FUMA software receiving a slightly amended input file with a (now) complete summary stats file. This resulted in slight changes to the outcome of tissue enrichment analysis.

- Table 1 – rs7765476 replaced with rs72828533

Both of these SNPs are in the same locus and also have the exact same p-value for association with sleep duration. rs72828533 has a slightly larger (in magnitude) beta coefficient.

Minor issues:

1. Figure 1 – what are the *'s indicating?

Authors: The bottom right hand side of Figure 1 contains text stating “* denotes same locus” i.e. rs7502280 (sleep duration) and rs2696625 (overall activity) are both in the same locus.

2. Table 1 and Suppl. Data 2 – in Table 1 rs7765476 has been replaced with rs72828533 but rs7765476 is still in Suppl. Data 2.

Authors: Thank you for noting this, we have now updated Suppl. Data 2 (see attached).

3. Suppl. Data 3 – there are two worksheets in this file – is this intended?

Authors: Thank you for noting this, we have now cleaned Suppl. Data 3 (see attached).

4. I am confused by the statement as it now seems contradictory with respect to whether additional loci were found or not: “Gene-based analysis using FUMA28, with input SNPs mapped by position to 18,232 protein-coding genes, implicated an additional locus for walking (near HP1BP3) and moderate activity (near KCNA6) (Supplementary Data 4). However, no further loci were found to those already identified in the SNP association analysis.”

Authors: Following consideration, we still remain satisfied that the two aforementioned sentences do not contradict each other.

5. Supplementary Table 12 – typo ‘accelerometer’ in first yellow box

Authors: Thank you for noting this, we have now updated Suppl. Data 12 (see attached).